# Factors that influence vaccination decision-making among pregnant women: A systematic review and meta-analysis

Eliz Kilich[1‡]*, Sara Dada[1‡], Mark R. Francis[1], John Tazare[2], R. Matthew Chico[3], Pauline Paterson[1], Heidi J. Larson[1,4]

1 Department of Infectious Diseases Epidemiology, Faculty of Epidemiology and Population Health, London School of Hygiene & Topical Medicine, London, United Kingdom, 2 Department of Medical Statistics, Faculty of Epidemiology and Population Health, London School of Hygiene & Topical Medicine, London, United Kingdom, 3 Department of Disease Control, Faculty of Infectious and Tropical Diseases, London School of Hygiene & Topical Medicine, London, United Kingdom, 4 Institute of Health Metrics and Evaluation, University of Washington, Seattle, Washington, United States of America

‡ EK and SD are joint first authors.
* eliz.kilich@lshtm.ac.uk

**Data Availability Statement:** All relevant data are within the manuscript and its Supporting Information files. Any additional data required will be submitted on the Editor's request.

## Abstract

### Background

The most important factor influencing maternal vaccination uptake is healthcare professional (HCP) recommendation. However, where data are available, one-third of pregnant women remain unvaccinated despite receiving a recommendation. Therefore, it is essential to understand the significance of other factors and distinguish between vaccines administered routinely and during outbreaks. This is the first systematic review and meta-analysis (PROSPERO: CRD 42019118299) to examine the strength of the relationships between identified factors and maternal vaccination uptake.

### Methods

We searched MEDLINE, Embase Classic & Embase, PsycINFO, CINAHL Plus, Web of Science, IBSS, LILACS, AfricaWideInfo, IMEMR, and Global Health databases for studies reporting factors that influence maternal vaccination. We used random-effects models to calculate pooled odds ratios (OR) of being vaccinated by vaccine type.

### Findings

We screened 17,236 articles and identified 120 studies from 30 countries for inclusion. Of these, 49 studies were eligible for meta-analysis. The odds of receiving a pertussis or influenza vaccination were ten to twelve-times higher among pregnant women who received a recommendation from HCPs. During the 2009 influenza pandemic an HCP recommendation increased the odds of antenatal H1N1 vaccine uptake six times (OR 6.76, 95% CI 3.12–14.64, $I^2$ = 92.00%). Believing there was potential for vaccine-induced harm had a negative influence on seasonal (OR 0.22, 95% CI 0.11–0.44 $I^2$ = 84.00%) and pandemic influenza vaccine uptake (OR 0.16, 95% CI 0.09–0.29, $I^2$ = 89.48%), reducing the odds of being

**Funding:** This research has been funded by a grant from GlaxoSmithKline to support research on maternal vaccination. The funders had no role in study design, data collection and analysis, decision to publish, or preparation of the manuscript.

**Competing interests:** HL's research group (HL, PP, MF, EK, SD) has received funds from GlaxoSmithKline and Merck (commercial funders). This research has been funded by a grant from GlaxoSmithKline (commercial funder) to support research on maternal vaccination. This does not alter our adherence to PLOS ONE policies on sharing data and materials. HL has also served on the Merck Vaccines Strategic Advisory Board 2014-2016. None of the other authors have conflicts of interest to declare.

vaccinated five-fold. Combined with our qualitative analysis the relationship between the belief in substantial disease risk and maternal seasonal and pandemic influenza vaccination uptake was limited.

## Conclusions

The effect of an HCP recommendation during an outbreak, whilst still powerful, may be muted by other factors. This requires further research, particularly when vaccines are novel. Public health campaigns which centre on the protectiveness and safety of a maternal vaccine rather than disease threat alone may prove beneficial.

## Introduction

Maternal vaccination aims to reduce maternal and neonatal morbidity and mortality caused by infection [1]. The World Health Organisation (WHO) recommends the inactivated influenza, tetanus-toxoid-containing vaccine (TTCV), and combined tetanus, diphtheria, and acellular pertussis (Tdap) vaccines for pregnant women in settings where the disease burden is known [2]. Historically, maternal tetanus vaccination was limited to areas of significant transmission. In areas where there is ongoing maternal to neonatal transmission of tetanus, two doses of TTCV (preferably Tetanus-diphtheria) are recommended in pregnancy in addition to Tdap or DTaP (for pertussis) and seasonal influenza vaccines.[2] Pertussis vaccination was limited to childhood, however the resurgence of pertussis during outbreaks that disproportionately affected younger infants led to national policy changes between 2011 and 2015 in countries such as the United Kingdom and the United States, that introduced routine maternal pertussis vaccination.[2,3] Similarly, the widespread influenza immunisation programs during the 2009 H1N1 pandemic resulted in public health bodies particularly in Europe, the United States and Australia introducing guidance to implement recommendations for routine antenatal seasonal influenza vaccination during the subsequent decade. The United States Healthy People 2020 campaign sets a target to achieve influenza vaccination coverage of 80% among pregnant women [4]. Suboptimal maternal vaccination coverage (estimated between 0–70%) of seasonal influenza and pertussis vaccines globally represents a missed opportunity to improve maternal and neonatal health [3–6]. Understanding the features that contribute to reduced uptake of vaccines used in outbreaks is also of particular importance given the increased morbidity and mortality seen with infections contracted during the vulnerable period of pregnancy [7].

In the last decade, the World Health Organisation has declared multiple Public Health Emergencies of International Concern for diseases including outbreaks of the Ebola virus (West Africa, North Kivu), Zika virus, and the novel Coronavirus (Wuhan) (COVID-19) [8, 9]. Ebola and Zika are known to cause significant morbidity and mortality if contracted during pregnancy, whereas the effect of the COVID-19 is unknown [10]. A vaccination strategy has been developed for Ebola, and vaccine research is underway for Zika virus and COVID-19. The concern of disease risk may be amplified during an outbreak, but concerns about using a novel vaccine may also be enhanced. It is important to identify factors that appear to affect antenatal vaccine uptake during routine use (pertussis and influenza) versus vaccinations recommended during an outbreak setting (H1N1 influenza) to help prepare for future outbreaks.

Understanding the influence of personal beliefs and experiences on maternal vaccination uptake is key to designing, testing and deploying interventions that are tailored to improve

vaccine acceptance and coverage in routine and outbreak settings. Researchers have investigated the underlying reasons for low coverage using surveys, focus group discussions, and in-depth interviews to explore the perceptions and experiences of pregnant women. Previous reviews have established a narrative of evidence that suggests a broad range of factors (vaccine cost, accessibility, maternal knowledge, social influences, context, healthcare professional (HCP) recommendation and the perception of risks and benefits) all contribute to vaccine uptake. Consensus within the field and across four prior literature reviews indicate that receiving a recommendation from an HCP for vaccination is the most important factor in maternal decision-making, irrespective of geographic or social context [11–15]. In general, there is limited data on maternal vaccination uptake and records of HCP recommendations at a national level. However, for the United States of America (USA), which monitors antenatal seasonal influenza and pertussis vaccination coverage, data suggest approximately one-third of women who receive an HCP recommendation for the vaccine will choose to remain unvaccinated [5]. In 2018, the Centers for Disease Control and Prevention (CDC) found that 79.3% of pregnant participants received a recommendation or an offer for Tdap vaccine, but 45.6% of them chose to remain unvaccinated [5]. For seasonal influenza, fewer women chose to vaccinate when recommended to do so; 81.1% received a recommendation or an offer yet 50.9% of pregnant women surveyed remained unvaccinated [5]. Understanding why women remain unvaccinated despite an HCP recommendation is key. We also sought to discriminate factors that influence specific vaccines since seasonal influenza vaccination coverage is lower than other routine vaccines (Tdap, tetanus) during pregnancy.

Prior literature and systematic reviews tend to characterize the factors influencing maternal vaccination decisions as either barriers or facilitators [11–15]. We sought to quantify the association between beliefs, attitudes and prior behaviours that influence maternal vaccination uptake. We selected the H1N1 Influenza vaccine, deployed globally and recommended to pregnant women during the pandemic of 2009, to be included alongside our analysis of other WHO routinely recommended vaccines, the pertussis and seasonal influenza vaccine. Thus, we performed a systematic review and meta-analysis of qualitative and quantitative literature to provide comprehensive evidence on the magnitude of effect that factors influence maternal vaccination decisions globally with the aim to inform policy makers, public health strategists and researchers involved in designing vaccine interventions to increase uptake.

## Materials and methods

### Search strategy and selection criteria

We conducted a systematic review of literature, unrestricted by language or location, to identify qualitative and quantitative studies that reported on the cognitive, psychological, and social factors associated with maternal vaccination among pregnant and recently pregnant women (within two years of birth). We searched MEDLINE, Embase Classic and Embase, PsycINFO, CINAHL Plus, Web of Science, IBSS, LILACS, AfricaWideInfo, IMEMR, and Global Health for studies published by 22 November 2018 (Appendix p3-10 in S2 File). Additional studies were identified by screening reference lists (EK, SD) of previous reviews and through suggestions by experts in the field.

Titles and abstracts were independently screened and agreed upon (EK, SD) for potential eligibility. A final arbiter (PP) resolved any conflicts of agreement on inclusion. We excluded pre-clinical research, behavioural intervention studies, and any research that exclusively examined experimental vaccines in pregnancy or sociodemographic variables (Appendix p11-15 in S2 File).

To be included in the meta-analysis, studies had to report an estimated odds ratio (OR (or could be calculated from raw data)) for the association between a specific factor and vaccination status (excluding intention to be vaccinated). Research groups from studies in which the data were unclear or had not been reported were contacted for clarification (Appendix p51 in S2 File).

### Data analysis

The data from included studies were extracted (EK, SD) and input into Microsoft Excel (Microsoft Corporation, Redmond, WA, USA) and a coding template was developed by authors to categorise factors influencing maternal vaccination uptake (Expanded Methodology Appendix p18-19 in S2 File details why established frameworks were not used). Coding into broad themes (e.g. accessibility and convenience) using grounded theory was completed independently (EK, SD) with NVivo 12 (QSR International, Melbourne, Australia) (Inter-rater reliability kappa score 0.76).

The quantitative studies were independently assessed (EK, MRF) for inclusion in the meta-analysis based on first cycle broad codes to capture data that could be synthesized (Appendix p38-40 in S2 File). Qualitative data underwent a second round of coding to identify specific patterns within the broad themes (Inter-rater reliability kappa score 0.88). A third round of coding (subdividing the first cycle codes) was conducted to ensure that only data that was directly comparable were included in each meta-analysis (Appendix p41 in S2 File). Twenty-three narrow definitions were agreed upon by two authors (EK, MRF) to ensure consistency of the data included (Appendix p42-43 in S2 File). These definitions were used to pool studies for specific vaccines (seasonal influenza, pandemic influenza, and pertussis) generating 31 separate meta-analyses (EK, MRF). Any discrepancies in data extraction were resolved by both authors.

Quality appraisal was performed (EK, SD) using checklists for cross-sectional, cohort, and qualitative research studies from the Joanna Briggs Institute quality assessment tools (Appendix p21-32 in S2 File).[16] Studies were ranked based on a framework developed by authors with an attributed quality score. Where authors disagreed on final point allocation, the arbiter (PP) intervened to resolve the disagreement. Quality analysis was not used to define inclusion or exclusion. However, pre-specified sensitivity analyses were performed investigating the robustness of results to the inclusion of only high-quality studies (Joanna Briggs Institute scores >10) (Appendix: p90, p92 in S2 File). We wished to conduct a sensitivity analysis assessing the robustness of results by Gross Domestic Product of countries included to assess the influence of geographic context.

When two or more studies reported ORs for a specific factor, random-effects models were used to calculate a summary OR [17] with heterogeneity assessed with $I^2$. Funnel plots were used to examine the potential for publication bias. Specific factors reported by only one study are summarised in the appendix (Appendix p93 in S2 File). A secondary analysis was performed to assess factors associated with intention to be vaccinated during pregnancy. All meta-analyses were conducted in Stata 15 (StataCorp, College Station, TX, USA) [18, 19].

The PRISMA checklist (Preferred Reporting Items for Systematic Review and Meta-Analysis checklist) (Appendix p16-17 in S2 File) [20] was used and the study was registered with PROSPERO (International Prospective Register of Systematic Reviews) (CRD42019118299). An expanded methodology can be found in the appendix (Appendix p19-21 in S2 File).

### Results

Of 17,236 articles screened, 120 were eligible for analysis (Fig 1). Table 1 summarises study characteristics, representing data from 73,251 pregnant or recently pregnant women and 30

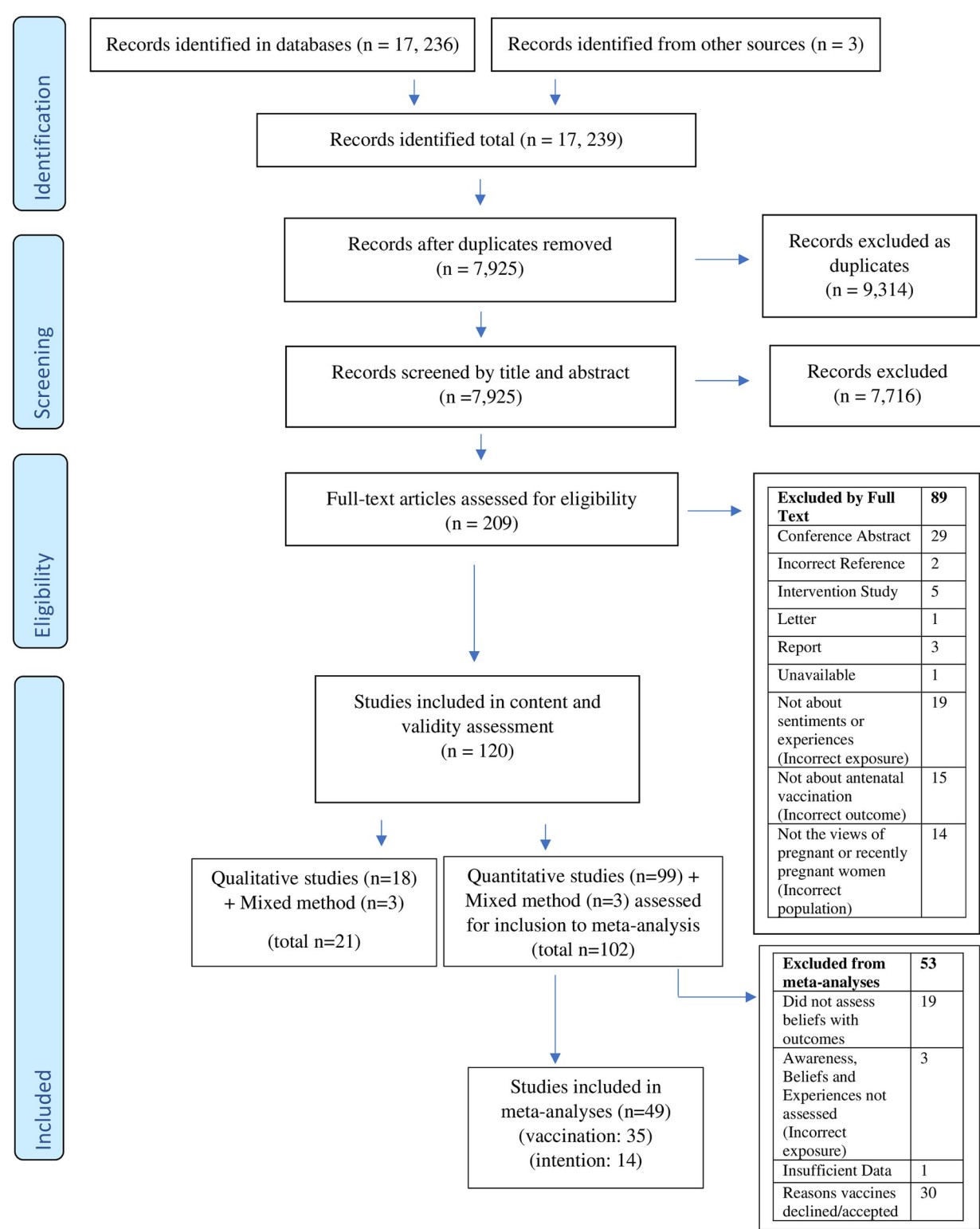

**Fig 1. PRISMA diagram of included and excluded studies.** Adapted from Moher D, Liberati A, Tetzlaff J, Altman DG, The PRISMA Group (2009). *Preferred Reporting Items for Systematic Reviews and Meta-Analyses: The PRISMA Statement. PLoS Med 6(7): e1000097. doi:10.1371/journal.pmed1000097.

**Table 1. Summary table of included studies 1996–2018.**

| Authors (year of publication) | | Year of study | Location | Study type | Vaccine | Number of participants | Sampling Method | Vaccine coverage of participants | Vaccine willingness (of unvaccinated participants) |
|---|---|---|---|---|---|---|---|---|---|
| Abasi et al. (2015) | [52] | 2013 | Iran | Questionnaire | Influenza | 384 | Convenience | 1.8%[A] | N/A |
| Agricola et al. (2016) | [96] | 2015 | Italy | Survey | Pertussis | 347 | Purposive | 1.7%[C] | 21%[C] |
| Arriola et al. (2018) | [21] | 2018 | Nicaragua | Survey | Influenza | 1303 | Purposive | 42%[A] | N/A |
| Ashfaq et al. (2017) | [113] | 2013 | Pakistan | Survey | Tetanus, Multiple | 46 | Convenience | 71.7%[D Or E] | N/A |
| Barrett et al. (2018) | [22] | 2016 | Ireland | Survey | Influenza | 198 | Convenience | 55.1%[A] 32.3%[B] | N/A |
| Barroso Pereira et al. (2013) | [67] | 2011 | Brazil | Interviews | Influenza (Pandemic) | 10 | Purposive | N/A | N/A |
| Beigi et al. (2009) | [114] | 2007 | USA | Questionnaire | Influenza (Pandemic) | 394 | Not specified | N/A | 15.7%[B] |
| Ben Natan et al. (2017) | [80] | 2017 | Israel | Survey | Pertussis | 200 | Convenience | N/A | 52%, 32% |
| Bettinger et al. (2016) | [68] | 2010 | Canada | Mixed Method | Influenza | 34 | Convenience | 45.6%[A] | 50%[A] |
| Bhaskar et al. (2012) | [115] | 2010 | India | Questionnaire | Influenza (Pandemic) | 140 | Random | 12.9%[B] | N/A |
| Blanchard-Rohner et al. (2012) | [48] | 2011 | Switzerland | Questionnaire | Influenza | 261 | Not specified | 18%[A] | N/A |
| Blondel et al. (2012) | [116] | 2010 | France | Survey | Influenza (Pandemic) | 14355 | Purposive | 29.3%[B] | N/A |
| Boedeker et al. (2014) | [23] | 2013 | Germany | Questionnaire | Influenza | 1025 | Random | 15.9%[A] | N/A |
| Boedeker et al. (2015) | [47] | 2012–2014 | Germany | Cohort Study | Influenza | 838 | Purposive | 10.9%[A] | N/A |
| Campbell et al. (2015) | [117] | 2013 | UK | Survey | Pertussis | 1892 | Purposive | N/A | >90%[C] |
| Cassady et al. (2012) | [75] | 2010 | USA | Focus Groups | Influenza (Pandemic) | 12 | Convenience | ~50%[B] | N/A |
| Castro-Sanchez et al. (2018) | [118] | 2015–2016 | Spain | Survey | Influenza, Pertussis | 119 | Not specified | 52%[A] , 94%[C] | N/A |
| Celikel et al. (2014) | [119] | 2010 | Turkey | Questionnaire | Influenza, (Seasonal and Pandemic) Tetanus, Other | 196 | Convenience | 3.0%[A], 9.1%[B] 47%[D] 0.5%[F] | N/A |
| Chamberlain et al. (2015) | [120] | 2004–2011 | USA | Survey | Influenza | 8300 | Random | 35.8%[A] | N/A |
| Chamberlain et al. (2016) | [121] | 2012–2013 | USA | Survey | Influenza, Pertussis | 325 | Stratified random | 9%[A,B] | 34%[A] 44%[C] |
| Collins et al. (2014) | [63] | 2011–2012 | Australia | Interviews | Influenza, Pertussis | 17 | Purposive | 11.8%[G] | 41.2%[G] |
| D'Alessandro et al. (2018) | [122] | 2017–2018 | Italy | Survey | Influenza, Pertussis | 358 | Cluster | 1.4%[A] 0%[C] | 27.9%[G] |
| Dempsey et al. (2016) | [81] | 2014 | USA | Survey | Pertussis | 316 | Convenience | N/A | 82%[C] |
| Ding et al. (2011) | [123] | 2010 | USA | Survey | Influenza (Seasonal and Pandemic) | 244 | Random | 32.1%[A] 45.7%[B] | N/A |
| Ditsungneon et al. (2016) | [82] | 2012–2013 | Thailand | Survey | Influenza | 1031 | Convenience | 4%[A] | 42%[A] |
| Dlugacz et al. (2012) | [42] | 2010 | USA | Survey | Influenza (Pandemic) | 1325 | Convenience | 34.2%[B] | N/A |
| Donaldson et al. (2015) | [83] | 2013–2014 | UK | Survey | Pertussis | 200 | Convenience | 26%[C] | 47.5%[C] |
| Drees et al. (2012) | [43] | 2010 | USA | Survey | Influenza (Pandemic) | 307 | Convenience | 60%[A] 62%[B] | N/A |

*(Continued)*

**Table 1.** (Continued)

| Authors (year of publication) | | Year of study | Location | Study type | Vaccine | Number of participants | Sampling Method | Vaccine coverage of participants | Vaccine willingness (of unvaccinated participants) |
|---|---|---|---|---|---|---|---|---|---|
| Drees et al. (2013) | [124] | 2009–2011 | USA | Survey | Influenza (Seasonal and Pandemic) | 300 | Not specified | 55%[A] | N/A |
| Edmonds et al. (2011) | [125] | 2009 | USA | Survey | Influenza (Seasonal and Pandemic) | 173 | Convenience | N/A | 63%[B] |
| Eppes et al. (2013) | [49] | 2009–2010 | USA | Survey | Influenza (Seasonal and Pandemic) | 88 | Convenience | 69%[A] 67%[B] | N/A |
| Fabry et al. (2011) | [54] | 2010 | Canada | Survey | Influenza (Pandemic) | 250 | Convenience | 76.4%[B] | N/A |
| Fisher et al. (2011) | [126] | 2009–2010 | USA | Questionnaire | Influenza (Seasonal and Pandemic) | 813 | Not specified | 34%[A] 63%[A] 54%[B] | N/A |
| Fleming et al. (2018) | [77] | 2015–2016 | El Salvador | Mixed Method | Influenza (Seasonal) | 117 | Convenience | 90%[A] | N/A |
| Fridman et al. (2011) | [53] | 2009 | USA | Questionnaire | Influenza (Pandemic) | 212 | Convenience | 25.5%[B] | N/A |
| Gaudelus et al. (2016) | [127] | 2016 | France | Questionnaire | Influenza, Pertussis | 300 | Quota | N/A | 74%[G] |
| Gauld et al. (2016) | [58] | 2014 | New Zealand | Interviews | Pertussis | 37 | Purposive | 46%[C] | N/A |
| Goldfarb et al. (2011) | [128] | 2010 | USA | Survey | Influenza (Seasonal and Pandemic) | 370 | Convenience | 81%[A+B] 7%[A] 4.7%[B] | N/A |
| Gorman et al. (2012) | [24] | 2010–2011 | USA/Canada | Survey | Influenza | 199 | Not specified | N/A | N/A |
| Gul et al. (2016) | [129] | 2015 | Pakistan | Questionnaire | Tetanus | 500 | Convenience | 73%[D Or E] | N/A |
| Hallisey et al. (2018) | [130] | 2015–2016 | Ireland | Questionnaire | Influenza, Pertussis | 88 | Purposive | 67%[C] 40%[A] | N/A |
| Halperin et al. (2014) | [131] | 2005–2006, 2011 | Canada | Survey | Influenza | 662 | Not specified | 67%[B] | N/A |
| Hasnain et al. (2007) | [132] | 2003–2004 | Pakistan | Questionnaire | Tetanus | 362 | Random | 87%[E] | N/A |
| Hassan et al. (2016) | [133] | 2015 | Egypt | Questionnaire | Tetanus | 277 | Convenience | 60.6%[D Or E] | N/A |
| Hayles et al. (2015) | [84] | 2013 | Australia | Survey | Influenza, Pertussis | 381 | Random | 39.1%[A] 8.7%[B] | 80.2%[B] |
| Healy et al. (2015) | [134] | 2013–2014 | USA | Survey | Influenza, Pertussis | 796 | Convenience | N/A | 69.3%[A] 51.6%[B] |
| Henninger et al. (2013) | [26] | 2010–2011 | USA | Survey | Influenza | 552 | Not specified | 46%[A] | N/A |
| Henninger et al. (2015) | [25] | 2010–2011 | USA | Cohort Study | Influenza | 1105 | Not specified | 63%[A] | N/A |
| Hill et al. (2018) | [135] | 2013 | New Zealand | Survey | Pertussis | 596 | Random | 74%[C] | N/A |
| Honarvar et al. (2012) | [136] | 2010–2011 | Iran | Questionnaire | Influenza (Seasonal and Pandemic) | 416 | Convenience | 6%[A, B] | N/A |
| Hu et al. (2017) | [85] | 2014 | China | Questionnaire | Influenza | 1252 | Convenience | 76.28%[A] | N/A |
| Jadoon et al. (2017) | [137] | | Pakistan | Questionnaire | Tetanus | ~210 | Cluster | 93%[D + E] 2%[D] | N/A |
| Kang et al. (2015) | [138] | 2011 | Korea | Survey | Influenza | 700 | Convenience | 27.3%[A] 19.3%[B] | N/A |
| Kay et al. (2012) | [44] | 2009–2010 | USA | Survey | Influenza (Seasonal and Pandemic) | 420 | Purposive | 70.9%[A] 76.9[b] | N/A |
| Kfouri et al. (2013) | [57] | 2010 | Brazil | Questionnaire | Influenza | 300 | Convenience | 95.7%[A] | 69.2%[A] (Had They Been Informed) |
| Khan et al. (2015) | [86] | 2013 | Pakistan | Questionnaire | Influenza | 283 | Convenience | N/A | 84.45%[A] |
| Kharbanda et al. (2011) | [64] | 2010 | USA | Focus Groups | Influenza | 40 | Convenience | 48%[A and/or B] | N/A |
| Kouassi et al. (2012) | [139] | 2010 | Ivory Coast | Survey | Influenza (Pandemic) | 411 | Random | N/A | 69.8%[B] |
| Koul et al. (2014) | [140] | 2012–2013 | India | Questionnaire | Influenza | 1000 | Convenience | 0%[A] | 100%[A] (Had They Been Rec & Informed Of Safety) |

(*Continued*)

Table 1. (Continued)

| Authors (year of publication) | | Year of study | Location | Study type | Vaccine | Number of participants | Sampling Method | Vaccine coverage of participants | Vaccine willingness (of unvaccinated participants) |
|---|---|---|---|---|---|---|---|---|---|
| Krishnaswamy et al. (2018) | [27] | 2016 | Australia | Survey | Influenza, Pertussis | 537 | Convenience | N/A | 57%[A], 63%[C] |
| Kriss et al. (2018) | [141] | 2018 | USA | Survey | Pertussis | 486 | Random | 40.7%[C] | N/A |
| Larson Williams et al. (2018) | [69] | 2016 | Zambia | Focus Groups | Pertussis | 50 | Purposive | 100%[D] | N/A |
| Lau et al. (2010) | [28] | 2005–2006 | Hong Kong | Questionnaire | Influenza | 568 | Convenience | 3.9%[A] | 33%[A] |
| Lohiniva et al. (2014) | [70] | 2009–2010 | Morocco | Focus Groups & Interviews | Influenza | 123 | Purposive | 54.5%[A] | N/A |
| Lohm et al. (2014) | [71] | 2011–2012 | Australia & Scotland | Focus Groups & Interviews | Influenza (Pandemic) | 14 | Purposive | N/A | N/A |
| Lotter et al. (2018) | [29] | 2015 | Australia | Survey | Influenza, Pertussis | 100 | Random | 60.0%[A] And 64.5%[C] | N/A |
| Loubet et al. (2016) | [30] | 2014–2015 | France | Questionnaire | Influenza | 153 | Convenience | 26%[A] | N/A |
| Lynch et al. (2012) | [74] | 2009 | USA | Focus Groups | Influenza (Pandemic) | 144 | Not specified | N/A | 48.5%[B] |
| MacDougall et al. (2016) | [142] | 2016 | Canada | Survey | Pertussis | 346 | Random | N/A | 89%[C] |
| Maher et al. (2013) | [31] | 2012 | Australia | Survey | Influenza | 462 | Random | 25%[A] | N/A |
| Maisa et al. (2018) | [61] | 2017 | UK: Northern Ireland | Focus Groups & Interviews | Influenza, Pertussis | 16 | Opportunistic | N/A | N/A |
| Mak et al. (2015) | [32] | 2012–2013 | Australia | Survey | Influenza, Pertussis | 831 | Random | 60.6%[A], 71.0%[C], 54.5%[A, C] | N/A |
| Mak et al. (2018) | [33] | 2015 | Australia | Survey | Influenza, | 424 | Random | 33.5%[A] | N/A |
| Marsh et al. (2014) | [65] | 2011–2012 | USA | Interviews | Influenza | 21 | Purposive | N/A | 95%[A] |
| de Mattos et al. (2003) | [143] | 1996 | Brazil | Questionnaire | Tetanus | 430 | Random | 24.20% | N/A |
| Maurici et al. (2016) | [34] | 2013 | Italy | Survey | Influenza | 309 | Purposive | 0%[A] | N/A |
| Mayet et al. (2017) | [144] | 2013 | Saudi Arabia | Questionnaire | Influenza | 998 | Purposive | 18.1%[A] | 74.2%[A] |
| McCarthy et al. (2012) | [145] | 2010–2011 | Australia | Questionnaire | Influenza | 199 | Not specified | 30–40%[A] | N/A |
| McCarthy et al. (2015) | [146] | 2010–2014 | Australia | Questionnaire | Influenza | 1086 | Not specified | 42.3%[A] | N/A |
| McQuaid et al. (2016) | [72] | 2014 | UK | Focus Groups | General | 14 | Purposive | N/A | N/A |
| McQuaid et al. (2018) | [147] | 2014–2015 | UK | Questionnaire | General | 269 | Purposive | 68% | N/A |
| Meharry et al. (2013) | [73] | 2010 | USA | Interviews | Influenza (seasonal and pandemic) | 60 | Purposive | 51.70%[A] | N/A |
| Mitra & Manna (1997) | [148] | 1997 | India | Survey | Tetanus | 100 | Convenience | N/A | N/A |
| Mohammed et al. (2018) | [35] | 2014–2016 | Australia | Survey | Influenza Pertussis | 180 | Not specified | 76%[A] 81%[C] | N/A |
| Napolitano et al. (2017) | [149] | 2015–2016 | Italy | Survey | Influenza | 372 | Random | 9.7%[A] | 21.4%[A] |
| O'Grady et al. (2015) | [56] | 2014 | Australia | Mixed Method | Influenza | 53 | Convenience | 17%[A] | 53%[A] |
| O'Shea et al. (2018) | [59] | 2016 | Ireland | Interviews | Influenza Pertussis | 17 | Purposive | 76.4%[A] 52.9%[C] | N/A |
| Og Son et al. (2014) | [50] | 2013 | South Korea | Questionnaire | Influenza | 218 | Convenience | 48.60% | N/A |

(Continued)

**Table 1.** (Continued)

| Authors (year of publication) | Year of study | Location | Study type | Vaccine | Number of participants | Sampling Method | Vaccine coverage of participants | Vaccine willingness (of unvaccinated participants) |
|---|---|---|---|---|---|---|---|---|
| Ozer et al. (2010) [150] | 2009–2010 | Turkey | Survey | Influenza (Pandemic) | 314 | Not specified | 8.9%[B] | N/A |
| Ozkaya Parlakay et al. (2012) [151] | 2009 | Turkey | Questionnaire | Influenza (Pandemic) | 86 | Convenience | N/A | N/A |
| Puchalski et al. (2014) [87] | 2015 | USA | Survey | Influenza | 60 | Purposive | 23.4%[B] | N/A |
| Regan et al. (2016) [152] | 2012–2014 | Australia | Survey | Influenza | 2018 | Random | 22.9% - 41.4%[A] | N/A |
| Richun et al. (2018) [66] | 2015–2016 | China | Focus Groups & Interviews | Influenza | 108 | Convenience | 0% | N/A |
| Sakaguchi et al. (2011) [153] | 2009 | Canada | Questionnaire | Influenza (Seasonal and Pandemic) | 130 | Convenience | 80%[B] | 27.7%[A] |
| Schindler et al. (2012) [76] | 2011 | Switzerland | Interviews | Influenza | 29 | Maximal variation | 17.2%[A] | N/A |
| Siddiqui et al. (2017) [88] | 2013 | Pakistan | Survey | Pertussis | 283 | Convenience | 86%[C] | N/A |
| Silverman & Greif (2001) [41] | 2001 | USA | Survey | Influenza | 242 | Convenience | 7.85%[A] | 51%[A] (If Physician Rec) |
| Song et al. (2017) [36] | 2012–2014 | China | Survey | Influenza | 1673 | Not specified | 0%[A] | 31%[A] (If Physician Rec) |
| Stark et al. (2016) [154] | 2013–2015 | USA | Survey | Influenza | 984 | Not specified | 77.9%[A], 36.1%[A, Mr] | 6.8%[A] |
| Steelfisher et al. (2011) [45] | 2010 | USA | Survey | Influenza (Pandemic) | 514 | Random | 42%[B] | 8%[B] |
| Strassberg et al. (2018) [155] | 2014–2015 | USA | Questionnaire | Influenza, Pertussis | 338 | Convenience | 35.8%[C] | 70.7%[A], 40.5%[C], |
| Taksdal et al. (2013) [37] | 2012 | Australia | Survey | Influenza | 416 | Stratified random | 25%, 23%[Mr] | N/A |
| Tarrant et al. (2013) [46] | 2010 | Hong Kong | Questionnaire | Influenza (Pandemic) | 549 | Purposive | 4.9%[A] 6.2%[B] 2.2% [A, B] | N/A |
| Tong et al. (2008) [38] | 2003–2004 | Canada | Survey | Influenza | 185 | Purposive | 14%[A] | N/A |
| Tuells et al. (2018) [156] | 2014–2015 | Spain | Survey | Influenza | 934 | Random | 27.90% | 96.8%[A] |
| Ugezu et al. (2018) [157] | 2018 | Ireland | Cohort Study | Influenza, Pertussis | 113 | Convenience | 42.5%[A] 31%[B] | N/A |
| Van Lier et al. (2012) [55] | 2010 | The Netherlands | Survey | Influenza (Pandemic) | 2993 | Random | 46%[B] | 39%[B] |
| Varan et al. (2014) [95] | 2012 | Mexico | Survey | Pertussis | 387 | Convenience | 45%[A] 74%[D] | 57%[C] |
| Vila-Candel et al. (2016) [51] | 2014–2015 | Spain | Survey | Influenza | 200 | Random | 40.5%[A] | 18%[A] |
| White et al. (2010) [158] | 2010 | Australia | Questionnaire | Influenza (Pandemic) | 479 | Convenience | 6.9%[B] | N/A |
| Wilcox et al. (2018) [159] | 2017–2018 | UK | Questionnaire | Influenza, Pertussis | 314 | Not specified | 38%[A] 56%[C] | N/A |
| Wilcox et al. (2019) [160] | 2017–2018 | UK | Questionnaire | Influenza, Pertussis | 314 | Convenience | 38%[A] 56%[C] | 40%[A], 46%[C] |
| Wiley et al. (2015) [60] | 2011 | Australia | Interviews | Influenza, Pertussis | 815 | Maximal variation | N/A | N/A |
| Wiley et al. (2013) [39] | 2011 | Australia | Survey | Influenza | 815 | Non-random stratified | 27%[A] | N/A |
| Wiley et al. (2013) [161] | 2011 | Australia | Survey | Pertussis | 815 | Non-random stratified | Na | 80%[C] |
| Ymba & Perrey (2003) [162] |  | Ivory Coast | Questionnaire | Tetanus | 124 | Not specified | 77.1%[D or E] 15%[F] | N/A |
| Yudin et al. (2009) [163] | 2006 | Canada | Survey | Influenza | 58 | Convenience | N/A | N/A |

(Continued)

**Table 1.** (Continued)

| Authors (year of publication) | | Year of study | Location | Study type | Vaccine | Number of participants | Sampling Method | Vaccine coverage of participants | Vaccine willingness (of unvaccinated participants) |
|---|---|---|---|---|---|---|---|---|---|
| Yuen *et al.* (2013) | [40] | 2010–2011 | Hong Kong | Questionnaire | Influenza | 2822 | Convenience | 1.7%[A] | N/A |
| Yuen *et al.* (2016) | [62] | 2011 | Hong Kong | Interviews | Influenza | 32 | Purposive | 6.3%[A] | N/A |
| Yun & Xu (2010) | [164] | 2009 | China | Survey | Influenza | 215 | Random | n/a | 45.4% |

[a] = influenza,

[b] = pandemic influenza (H1N1),

[c] = pertussis,

[d] = tetanus (TT1),

[e] = tetanus (TT2),

[f] = hepatitis B,

[mr] = medical record,

[g] = general.

Questionnaire only = 35, Survey only = 58, Interviews only = 9, Focus Groups only = 5, Cohort Study = 3, Focus Groups & Interviews = 5, Surveys & Focus Groups = 2.

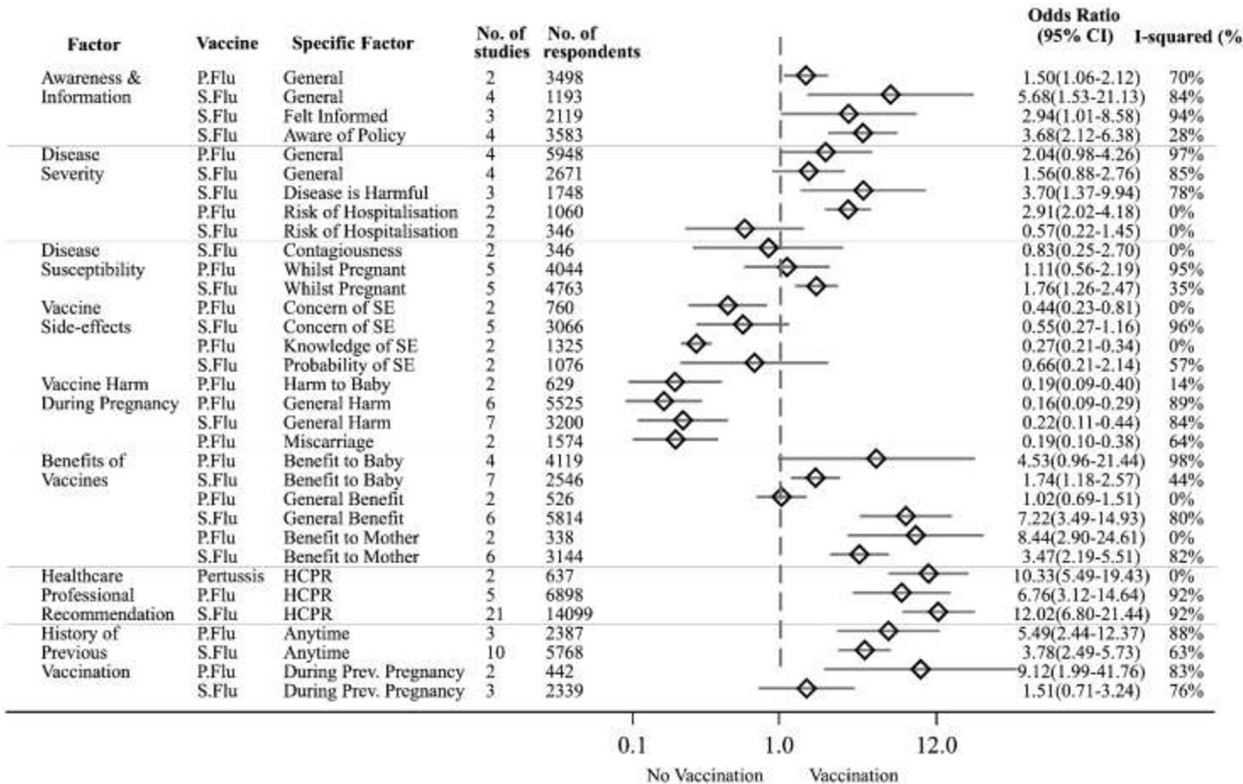

**Fig 2. Factors associated with maternal vaccine uptake—A summary forest plot.** Abbreviations. HCPR—healthcare professional recommendation, General—generally, P. Flu—pandemic influenza vaccine, SE—side effects, S. Flu—seasonal influenza vaccine.

countries (Appendix p33-37 in S2 File). The majority of studies were quantitative only (n = 99), then qualitative only (n = 18) with three studies using mixed methods (Table 1). Studies were predominantly from the USA (39 studies), Australia (22), and Canada (9). Seasonal influenza vaccine was the most commonly investigated vaccine, either independently or as part of a study of factors influencing the uptake of multiple vaccines (63% of studies n = 75), followed by vaccines against pertussis (27% n = 32), pandemic influenza (24% n = 29), tetanus (8% n = 9), and antenatal vaccines generally (2% n = 2).

We identified eight categories of factors that influence maternal vaccination across both qualitative and quantitative studies: accessibility and convenience (55 studies), personal values and lifestyle (43), awareness of information regarding the specific vaccine or disease of focus (90), social influences on vaccine use (109), emotions related to vaccination (85), perceptions of vaccine risk (110), perceptions of vaccine benefit (93), and personal vaccination history (80). From these eight categories, five could be synthesised quantitatively (Appendix p40-41 in S2 File). Results from all meta-analyses are presented in Fig 2. No data for tetanus vaccination were suitable for meta-analysis. A list of the most common barriers or facilitators cited in studies excluded from meta-analysis is available in the appendix (Appendix p52-53 in S2 File). From the 21 qualitative studies, we identified 30 sub-categories of factors that appear to influence maternal vaccination decision-making (Table 2).

## Quantitative studies

For our primary analysis we conducted 33 meta-analyses which assessed the relationship between a specific belief or behaviour and maternal vaccination status (338–14,099 participants

**Table 2. Qualitative study themes.**

| | Barroso Pereira et al. 2013. [67] | Bettinger et al. 2016. [68] | Cassady et al. 2012. [75] | Collins et al. 2014. [63] | Fleming et al. 2018. [77] | Gauld et al. 2016. [58] | Kharbanda et al. 2011. [64] | Larson Williams et al. 2018. [69] | Lohiniva et al. 2014.[70] | Lohm et al. 2014. [71] | Lynch et al. 2012. [74] | Maisa et al. 2018. [61] | Marsh et al. 2014. [65] | McQuaid et al. 2016. [72] | Meharry et al. 2013. [73] | O'Grady et al. 2015. [56] | O'Shea et al. 2018. [59] | Richum et al. 2018. [66] | Schindler et al. 2012. [76] | Wiley et al. 2015. [60] | Yuen et al. 2016. [62] | No. of studies |
|---|---|---|---|---|---|---|---|---|---|---|---|---|---|---|---|---|---|---|---|---|---|---|
| **Country** | Brazil | Canada | Latino, USA | Aus | El Salv | France | USA | Zambia | Morocco | Aus, Scotl | USA | UK | USA | UK | USA | Aus | Ireland | China | Switz | Aus | Hong Kong | |
| **Vaccine studied** | P.Flu | S. Flu | P.Flu | Gen | P. Flu | Multi | S.Flu | Pertu | S. Flu | P. Flu | P. Flu | Mult | S. Flu | Gen | S. Flu | S. Flu | Multi | S Flu | S Flu | Multi | S Flu | |
| **Perception of Risk** | | | | | | | | | | | | | | | | | | | | | | |
| Disease severity | x | x | x | | | | x | x | x | x | x | x | x | | x | x | | x | x | x | x | 16 |
| Disease susceptibility | x | x | | x | | x | x | | x | x | x | | | | x | | | x | | | x | 10 |
| Risk of perceived vaccine harm | x | x | x | x | | x | x | x | x | x | x | x | x | x | x | | x | x | x | x | x | 19 |
| Risk of known vaccine side effects | x | | x | | x | x | | | | x | x | x | x | | x | x | | | x | | | 11 |
| Vaccine perceived as safe | | x | | | | | | | | | x | | x | | | x | | | | | | 4 |
| **Perception of Vaccine Benefits** | | | | | | | | | | | | | | | | | | | | | | |
| Vaccine protects pregnant woman | x | x | | | x | | x | x | x | | x | | | | x | x | | | | | | 9 |
| Vaccine protects foetus | x | | | | | | | | | | x | | | | x | x | | | | | | 4 |
| Vaccine protects baby | | x | | | x | | x | x | x | | x | | x | | x | x | | | | | x | 10 |
| Not protective/useful | x | x | x | | | | x | | x | | | x | x | | | | x | x | x | | x | 11 |
| | 13 | 16 | 10 | 13 | 8 | 11 | 14 | 11 | 16 | 12 | 10 | 16 | 16 | 7 | 20 | 14 | 10 | 13 | 12 | 11 | 13 | |

Abbreviations: P. Flu = pandemic influenza vaccine, S Flu = seasonal influenza vaccine, Pertu = pertussis vaccine, Mult–multiple vaccine, General = antenatal vaccines generally, Aus = Australia, El Salv = El Salvador, Scotl = Scotland, Switz = Switzerland, UK = United Kingdom, USA = United States of America.

(average 2,955) included in each meta-analysis)) (Appendix p55-89 in S2 File for individual meta-analyses and summary table) (Fig 2). For our secondary analysis we conducted 15 meta-analyses assessing the relationship between a specific belief or behaviour and maternal vaccination intentions, rather than prior behaviour (531–2,215 participants (average 1,344) included in each meta-analysis)) (Appendix p91 in S2 File). The majority of studies with quantitative results for the pertussis vaccine had investigated intention to be vaccinated rather than actual vaccination status. Pregnant women who had received an HCP recommendation had 12-times higher odds of accepting seasonal influenza vaccination (OR 12.02, 95% CI 6.80–21.23, 21 studies, 14,099 women) [21–41] and 10-times greater odds of accepting pertussis vaccine (OR 10.33, 95% CI 5.49–19.43, 2 studies, 637 women) [27, 29] compared to those who had not received recommendations. For pandemic vaccine the recommendation increased the odds of antenatal H1N1 vaccine uptake by six times (OR 6.76, 95% CI 3.12–14.64, 5 studies, 6898women) [42–46]. The odds of pregnant women receiving season influenza vaccination were five-times higher if they had general information about the vaccine (OR 5.68, 95% CI 1.53–21.13,4 studies, 1193 women) [22, 27, 31, 47]. Similarly, the odds of being vaccinated were three-times higher (OR 3.68, 95% CI 2.12–6.38, 4 studies, 3583 women) [37, 40, 48, 49] among pregnant women who knew there was a national vaccination policy in place versus women who were unaware.

Prior vaccination history was influential in subsequent maternal vaccination decisions. The odds of being vaccinated against seasonal influenza were three-times greater (OR 3.78, 95% CI 2.49–5.73, 10 studies, 5,768 women) [23, 24, 27, 38, 40, 41, 48–51] and five-times greater for vaccination against pandemic influenza (OR 5.49, 95% CI 2.44–12.37, 3 studies, 2,387 women) [42, 45, 46, 52] if pregnant women had received vaccines as adults outside of pregnancy. Pregnant women who received a seasonal influenza vaccination during a prior pregnancy had nine-times higher odds of accepting a pandemic influenza vaccine in their current pregnancy than those who did not vaccinate in a prior pregnancy (OR 9.12, 95% CI 1.99–41.76, 2 studies, 442 women) [43, 53]. There was no evidence to suggest an association between prior maternal vaccination and season influenza vaccinations (OR 1.51, 95% CI 0.71–3.24, 3 studies, 2,339 women) [21, 22, 47].

The odds of accepting the pandemic influenza vaccine were six-times lower when women perceived it as unsafe in pregnancy (OR 0.16, 95% CI 0.09–0.29, 6 studies, 5,525 women) [42, 44–46, 54, 55]. Similarly, the odds of accepting the seasonal influenza vaccine were 86% lower when women believed receiving the vaccine during pregnancy was unsafe (OR 0.22, 95% CI 0.11–0.44, 7 studies, 3,200 women) [24, 25, 31, 37, 39, 48, 56]. Additionally, perceiving that the pandemic influenza vaccine caused harms such as birth defects (OR 0.19, 95% CI 0.09–0.40, 2 studies, 629 women) [46, 49] or miscarriage (OR 0.19, 95% CI 0.10–0.38, 2 studies, 1,574 women) [42, 46] were both associated with a five-times lower odds of vaccination. |Having concerns about pandemic influenza vaccine side-effects in general (OR 0.44, 95% CI 0.23–0.81, 2 studies, 760 women) [46, 53] was associated with two-times lower odds of vaccination; having knowledge of specific pandemic influenza vaccine side-effects (defined as a awareness of a known adverse reaction as outlined by the drug company label insert, e.g. fever.) (OR 0.27, 95% CI 0.21–0.34, 2 studies, 1,325 women) [42, 54] was associated with three-times lower odds of vaccination.

In contrast, the odds of being vaccinated were eight-times higher when pregnant women believed that the pandemic influenza vaccine benefits the mother (OR 8.44, 95% CI 2.90–24.61, 2 studies, 338 women) [49, 54]. The odds of accepting the seasonal influenza vaccine were seven-times greater when pregnant women perceived the vaccine as generally effective (OR 7.22, 95% CI 3.49–14.93, 6 studies, 5,814 women) [21, 24, 37, 39, 40, 57], three-times greater when they believed the vaccine benefits the mother (OR 3.47, 95% CI 2.19–5.51, 6

studies, 3,144 women) [23, 25, 31, 37, 49, 56], and almost two-times greater when they believed the vaccine benefits their baby (OR 1.74, 95% CI 1.18–2.57, 7 studies, 2546 women) [25, 31, 37, 38, 41, 49, 56].

There was insufficient evidence on the influence of perceived susceptibility to pandemic influenza during pregnancy on pandemic influenza vaccination uptake (OR 1.11, 95% CI 0.56–2.19, 5 studies, 4,044 women) [45, 49, 53–55]. However, pregnant women who felt they were susceptible to seasonal influenza had almost two-times higher odds of vaccination than those who did not feel susceptible to contracting seasonal influenza (OR 1.76, 95% CI 1.26–2.47, 5 studies, 4,763 women) [24–26, 40, 49]. There was inconclusive evidence to support a similar association between perceptions of the severity of pandemic (OR 2.04, 95% CI 0.98–4.26, 4 studies, 5,948 women) [42, 44, 53, 54] or seasonal influenza (OR 1.56, 95% CI 0.88–2.76, 4 studies, 2,671 women) [24–26, 39] with vaccination status. However, pregnant women who believed seasonal influenza could be harmful to their pregnancy or baby had four-times greater odds of being vaccinated than those who did not believe seasonal influenza could affect their pregnancy or baby (OR 3.70, 95% CI 1.37–9.94, 3 studies, 1,748 women) [23, 31, 48].

When the number of studies included in the meta-analysis exceeded seven, funnel plots were used to assess the potential for publication bias (Appendix p54 in S2 File). Although based on a small number of studies, there was incomplete agreement between the primary and secondary analyses (intention to be vaccinated meta-analysis results are provided in Appendix p91-92 in S2 File). Sensitivity analyses including studies with Joanna Briggs Institute scores >10 were conducted for both vaccination status and intention to be vaccinated outcomes (Appendix: p90, p92 in S2 File). Whilst results were generally consistent, differences were difficult to interpret due to the low number of higher-quality studies.

## Qualitative studies

All qualitative studies reported on the perceived effect of HCP influence on decision-making, and to a lesser extent the influence of other social networks or the Internet. Often an offer (or lack of an offer) of vaccination during an antenatal visit was a key factor in final behaviour [58–60]. Participants also expressed willingness to receive information from HCPs, but were disappointed with a perceived overuse of leaflets to convey information in lieu of direct conversation with an HCP [58, 61, 62]. Other studies reported that some pregnant women sought vaccination information through media or the Internet, but these avenues were not regarded as the most reliable for accurate information [62–66].

Almost all qualitative studies indicated that being aware of maternal vaccination and/or the respective disease, regardless of information source, was key to receiving the vaccine but rarely sufficient (17 studies) [56, 58–73]. Furthermore, 16 qualitative studies highlighted an information gap specific to knowledge of vaccines during pregnancy [56, 58–71, 73], reflecting a general lack of awareness among pregnant women of maternal vaccine recommendations and benefits.

Qualitative studies identified a number of additional health concerns about antenatal vaccines such as narcolepsy [58], infertility [70], autism [65], and unknown risks [61, 62, 68, 74–76] in addition to concerns of birth defects [63, 65, 68] and miscarriage [62–64, 68, 70] (Table 2). Specific side-effect concerns included vomiting, fever, body aches, soreness, fainting, seizures, illness, and unknown short and long-term side-effects [56, 58, 61, 65, 67, 71, 73–77]. Whilst the benefits of vaccines were mentioned in 16 qualitative studies, 11 of these studies reported that there was also doubt and uncertainty around the usefulness or the efficacy of vaccines in pregnancy [59, 61, 62, 64–66, 68, 70, 73, 75, 76].

In eight of the 17 qualitative studies that examined perceptions of disease severity, participants were unaware of the additional risks of influenza to pregnant women [56,62, 64, 66–68, 74, 75]. Qualitative studies also highlighted differences in participants' perceptions of severity for different diseases. For example, H1N1 or pandemic influenza was perceived as more severe than seasonal influenza [68, 74]. Additionally, pertussis was correctly seen primarily to present danger to infants, whereas influenza was viewed as a significant risk to the pregnant woman [60]. Whilst the disease risk was used as a contributing factor to final decision other factors were weighed against it.

Whilst factors such as convenience, personal values, and emotions related to vaccinations during pregnancy were not captured in our meta-analyses, they were highlighted in qualitative analyses. Several studies reported on vaccine availability [56, 59–62, 65, 70, 77], access [58, 73, 77], and competing priorities in pregnancy [56, 58, 60, 61, 73]. In some studies, participants may have accepted vaccines generally, but not during pregnancy [56, 66, 76]. Community rumours and cultural values also influenced views on vaccination among pregnant women [69, 70, 73, 75]. Additionally, several studies reported preferences for natural immunity or a healthy lifestyle during pregnancy as reasons to decline vaccinations [61, 62, 66, 68, 76]. Maternal vaccination decision-making was also associated with several emotions and sentiments including fear (13 studies) [61, 62, 64, 67–73, 75–77], worry or anxiety (8 studies) [56, 58, 61, 63, 67, 70, 72, 77], responsibility for pregnancy outcomes and culpability if something goes wrong (5 studies) [61, 68, 71, 73, 76], and uncertainty about risks associated with vaccination decisions (3 studies) [63, 75, 76]. Pregnant women feared the unknown [68, 70–72] the disease (particularly for pandemic influenza) [62, 68, 70, 73], vaccine harm or side-effects [61, 64, 67, 69, 70, 75, 76], vaccine safety [64, 73], and pain [52, 61,77]. One study reported that vaccinated and unvaccinated pregnant women expressed similar fears, but unvaccinated women often described their fears in more detail [70]. The fear of perceived vaccine harms (including the ideas of unknown risks for novel vaccines) were used to explain the rejection of maternal vaccination despite a connected fear of the disease it was aimed to protect against [59, 64–68, 70–73, 75].

## Discussion

Despite the challenges of synthesizing an extensive and varied body of research, we have been able to quantify the relative effect size for a large number of specific beliefs and behaviours around maternal vaccination uptake. Prior attempts to weight factors influencing maternal vaccination uptake have largely been confined to ranking the most commonly cited barriers or facilitators within studies, listing the latter as predictors. This approach is likely to conflate several individual factors which are important to designing better interventions aimed at increasing maternal vaccination acceptance and uptake.

Our major finding is that *vaccine-specific* factors and previous vaccination behaviour have a strong influence on antenatal vaccine uptake. *Disease-related* perceptions have a modest effect on final vaccination uptake. Beliefs that vaccine would benefit the mother or cause no harm to the pregnancy were associated with four-to-nine-times greater odds of vaccination-acceptance during pregnancy. Prior systematic reviews were unable to characterise the nature or strength of effect of vaccine safety concerns on maternal vaccination decisions. Lutz et al. described that 2.9 to 77% of pregnant women had safety concerns for their foetuses [13]. Wilson *et al.* reported that safety concerns were the most frequently cited barriers (64 of 155 studies) but the relationship of this concern to final vaccination uptake was not defined. The underlying vaccination status of pregnant women was unreported in this prior study, limiting the interpretation of this finding [11]. In our study, beliefs that vaccine could cause birth defects or

general harm in pregnancy served as strong deterrents to both seasonal and pandemic influenza vaccination (seasonal OR 0.22, 95% CI 0.11–0.44; pandemic OR 0.11, 95% CI 0.06–0.22). Similarly, perceptions of vaccine utility had a strong positive influence on uptake. For the seasonal influenza vaccine, perceiving the vaccine as beneficial in general was an important factor associated with pregnant women's vaccination status (OR 7.22, 95% CI 3.49–14.93). For pandemic influenza vaccination, despite the wide confidence intervals, our data suggest that perceptions that vaccine can protect pregnant women (OR 8.44 95% CI 2.90–24.61) is strongly associated with vaccine uptake.

Our study did not find clear evidence that a belief of susceptibility to pandemic or seasonal influenza was associated with increased maternal pandemic influenza vaccination (OR 1.11, 95% CI 0.56–2.19) or seasonal influenza (OR 1.76, 95% CI 1.26–2.47) vaccine uptake. There was some evidence to support an association between perceptions of the severity of pandemic influenza and pregnant women's vaccination status (OR 2.04, 95% CI 0.98–4.26) with the belief that pandemic influenza can result in hospitalisation increasing vaccine uptake three-fold (OR 2.91, 95% CI 2.02–4.18). We would recommend additional studies to explore the role of disease severity and susceptibility in greater detail to clarify their importance when other factors are present. For seasonal influenza, the data is inconclusive since women who believed that the disease could be harmful to their pregnancy or baby had four-times greater odds of being vaccinated than those who did not (OR 3.70, 95% CI 1.37–9.94) yet there was no evidence to suggest belief in the risk of the disease generally (OR 1.56, 95% CI 0.88–2.76) or its ability to result in hospitalisation (OR 0.57, 95% CI 0.22–1.45) were related to vaccine uptake. This was mirrored by our qualitative research which indicated that the influence of a belief in the severity and susceptibility to a disease does not in isolation determine vaccination decision [56, 58, 59, 63–71, 73–76]. This has important implications for public health communication strategies around maternal vaccination since campaigns, particularly during an epidemic or influenza outbreak, have centred around disease threat. Based on our findings we caution any communication approach which highlights only the threat of disease when publicising vaccination. We suggest this requires further review of the messaging strategies comparing those with and without explicit details of vaccine safety to the public and/or a discussion of disease threat with attention to language which might inadvertently promote fear [78]. The global communication strategies during the H1N1 2009 pandemic have been widely criticised as lacking an evidence-base and not appropriately targeting specific vulnerable groups [79]. We suggest that future research investigates disease-focused communication strategies versus vaccine-centred communication when discussing maternal vaccination to help prepare for future pandemics. Our study was conducted prior to the North Kivu Ebola vaccine deployment among pregnant women in 2019. This study precedes vaccine candidate deployment for SARS-CoV-2 with immediate implications for future studies analysing potential acceptance of a maternal vaccine and the associated communication strategy. This is an important area of investigation to analyse the factors that influence maternal uptake when the vaccine is still in an experimental part of outbreak control. Whilst the analysis of purely experimental vaccines was outside the remit of this work, we suggest further investigation into assessing the importance of factors identified in this review (including fear-conflict, anxiety and specific safety concerns) and their influence on uptake of vaccine during its developmental phase at the time of an outbreak.

Our findings also have potential implications for future study design. Many studies included in this systematic review and meta-analysis were designed using the framework of the Health Belief Model [24–26, 28, 39, 40, 53, 54, 60, 62–64, 69, 73, 80–88]. In brief, the Health Belief Model describes final vaccination acceptance or rejection based on the interacting beliefs of seriousness and susceptibility to the target disease of the vaccine, benefits of the intervention,

and barriers in order to predict health behaviour. Our study suggests that the model should be adapted to highlight the importance of the latter two categories in maternal vaccination behaviour predictions.

Consistent with the extensive body of evidence on this topic, an HCP recommendation for routine vaccinations (seasonal influenza and pertussis vaccination) was a very strong factor influencing maternal vaccine acceptance that is associated with ten-times greater odds of being vaccinated over those who did not receive an HCP recommendation (pertussis OR 10.33, 95% CI 5.49–19.43; seasonal influenza OR 12.02, 95% CI 6.80–21.44).

Although based on a small number of studies, our meta-analysis suggests that the influence of an HCP recommendation for pandemic influenza vaccination moderately-to-strongly influences uptake. Pandemic vaccine uptake was closely related to prior vaccination behaviour. Vaccination (with a different vaccine) during a prior pregnancy (OR 9.12 95% CI 1.99–41.76) had a strong influence on pandemic vaccine uptake. Interestingly, this did not appear as evident for receiving an antenatal seasonal influenza vaccine in a subsequent pregnancy (OR 1.51, 95% CI 0.71–3.24). This suggests a possibility that decision-making for seasonal influenza vaccines made in second and third pregnancies may not be consistent with the decisions in the first pregnancy [89]. Whilst studies have often included a sample of second- or third-time mothers, there is less extensive evidence of temporal changes in decision-making factors for maternal vaccines.

Whilst a general awareness about maternal vaccination did not increase the odds greatly of pandemic vaccine uptake, it appeared key for routine antenatal vaccines. Policy awareness was strongly associated with seasonal influenza vaccination uptake. National recommendations from the authoritative health bodies appear to carry weight in maternal vaccination decision-making at a population level [90]. This is important for the rollout of new maternal vaccines, as vaccinations not endorsed by national policy may be less accepted. Publicising such policies could improve trust in maternal vaccination programmes and facilitate improved uptake.

Qualitative findings from focus group discussions and in-depth interviews were generally consistent with the quantitative results: an unambiguous recommendation from an HCP to vaccinate against seasonal influenza or pertussis is key to pregnant women being vaccinated [58–66, 68, 72, 73, 76, 77]. It is difficult to draw conclusions about which specific HCP (e.g. obstetrician, general practitioner, and midwife) or service provider (e.g. community or hospital-based practitioners) is the most influential. Similar to the quantitative literature, qualitative studies have shown that a recommendation by an HCP was not always sufficient [26, 34, 36, 41, 45, 46, 62, 63, 67]. Reasons for refusal despite HCP recommendation from the qualitative analysis provide insights into the effects of fear, mistrust, and a feeling of accountability [61, 63, 67, 73, 75]. In the face of uncertainty about a vaccine, a guarded state prevails despite concern for disease risk [61, 68, 71, 73, 76]. This was most notably captured by Meharry *et al.* as, "...fear if I do (vaccinate), fear if I don't (vaccinate), and do nothing when I fear both" [73]. This analysis, combined with the finding of a very strong relationship between the belief of vaccine harm and reduced uptake indicates that the perceived risk of self-intervening (i.e. taking a vaccine) can be very powerful. This can overshadow the belief in environmental risk (e.g. contracting a severe disease). This suggests that mothers feel accountable for a perceived risk when choosing to vaccinate during pregnancy, which can result in inaction if the disease is also feared. Whilst it is essential that pregnant women are informed about the risks of the disease in order to be appropriately consented, the manner in which this is communicated should be evaluated. It appears that, in some cases, fear may be counter-productive. This has been seen in childhood vaccines: if parents already fear the vaccine, making them fear the disease leads to decision conflict, and hesitancy [91].

By including qualitative analysis, we were also able to unravel specific, participant-driven concerns that ranged from possible adverse events such as narcolepsy, infertility, and autism spectrum disorder as well as pregnancy-related concerns such as suspected risks of miscarriage, preterm birth, and birth defects. Since the non-pregnancy related health concerns occur rarely in the general population or require long-term follow up over decades, post-marketing surveillance studies are used to measure if there are any vaccine-related effects [92]. However, data on these specific concerns during pregnancy are often unavailable to general practitioners or midwives during counselling. We recommend that HCPs are given ready access to clear and concise language on the safety of vaccines during pregnancy. The CDC has launched an extensive response to the relationship between antenatal vaccines and Guillain Barre syndrome, autism, febrile seizures and sudden infant death syndrome (SIDS) [93–103]. However, discussions on narcolepsy are made in reference to childhood rather than prenatal vaccination. Additionally, there is limited availability of summarized reports for the public or general practitioners that synthesize the abundance of safety evidence on miscarriage, infertility, and birth defects. Health bodies should make this widely generated safety evidence more accessible to the public and to HCPs to facilitate uptake where concerns in practice arise [103, 104].

It is reassuring that our meta-analysis reinforces some of the existing evidence surrounding factors that influence maternal vaccine uptake. Attempting to quantify an effect size adds a useful summary measure. However, our study has a number of limitations that potentially impact inferences drawn from these data. The evidence-base exploring the factors determining antenatal vaccination decisions is extensive but of mixed quality, and synthesis of the results was complicated by the contextual and methodological heterogeneity between studies. A particular challenge was synthesizing questionnaire data which used variable phrasings and posed different assortments of questions limiting the volume of data that could be synthesised. Ninety-seven percent of quantitative studies pooled employed a cross-sectional design. We acknowledge that in some settings, where data are obtained using variable questionnaires, examining a different number of factors, pooling information may obtain inconsistent results. However, in practice, it is unclear how these differences are likely to influence the obtained summary estimates. This has highlighted the need for more standardised procedures for data collection and reporting for individual studies. This is of particular importance during outbreaks when research is delivered in a timely manner.

Whilst we intended to compare results from the meta-analyses with vaccination status and intention to be vaccinated as outcomes, data were insufficient to draw meaningful comparisons. All the data for vaccine intention is found in the Appendix 23 p91 in S2 File. This is important since the majority of data available for pertussis vaccination in pregnancy focused on beliefs associated with vaccination intentions only [81, 88, 105, 106]. Whilst previous literature has shown intention to vaccinate can be a proxy for actual vaccination status, this may not always be the case with maternal vaccination and additional research is needed [68, 107].

Our findings were largely consistent across countries; however, we recognize that the majority of data come from high-income settings where national vaccine policies exist and, therefore, the generalisability of these findings may be limited. We were unable to conduct a sensitivity analysis to stratify results by Gross Domestic Product of each country as there were too few studies included in each meta-analysis. Additionally, data from many studies relied on self-reported vaccination status and did not verify medical records or vaccine registry data which may have introduced a recall bias (reflected in the JBI quality assessment scoring). However, previous research in the field has indicated that this bias is unlikely to be substantial [108, 109]. A number of studies recruited participants using purposive or convenience sampling (Table 1). Thus, we applied the Joanna Briggs Institute (JBI) Critical Appraisal Tools (JBI) to assess potential biases among individual studies. Based on our results, we then performed a

pre-defined sensitivity analysis (of high quality/low quality) presented in the Appendices (Appendix 22, Appendix 24 p90, p92 in S2 File). We were unable to detect an influenced study quality on our pooled analyses. A final limitation is that the exposures in our meta-analyses were dichotomized, whereas in reality beliefs exist on a spectrum.

Vaccine refusal is undoubtedly multifactorial. However, our study has demonstrated that factors specific to the vaccine, perhaps more so than the disease are highly influential. Interventions recommended to improve maternal vaccination uptake have ranged from text reminders for prospective mothers to educational videos and motivational interviewing techniques for HCPs [110–112]. Based on the results of this review, interventions designed to impact maternal vaccine uptake should continue to encourage individualised HCP recommendations. Additionally, personalised counsel on the benefits and safety of a vaccine should emphasize the vaccine's protective effect on the pregnancy as well as discuss implications for foetal and childhood development. This is in contrast to traditional communication on disease threat in isolation. Readily accessible information that synthesizes the large body of evidence that may otherwise appear contradictory to the general public, will facilitate healthcare consultations in addressing pregnancy and long-term concerns, such as those identified in our review.

## Supporting information

**S1 Dataset.**
(XLSX)

**S1 File. PRISMA checklist.**
(DOCX)

**S2 File. Appendix.**
(DOCX)

## Acknowledgments

Tim Clayton (LSHTM, help with statistical queries), Kerrie Wiley (University of Sydney, data), Joseph Lau (Chinese University of Hong Kong, data), Meagan Kay (Division of Applied Sciences, CDC, Atlanta, Georgia, data), Dmitry Fridman (Department of Obstetrics and Gynecology, Maimonides Medical Center USA, data), Allison Chamberlain (Department of Epidemiology, Rollins School of Public Health, Emory University, USA, data), Alies van Lier (RIVM–Centrum Infectieziektebestrijding, data) Language Connect (for language translations), Daniel S. Epstein (Monash University, Australia, review of manuscript).

## Author Contributions

**Conceptualization:** Eliz Kilich.

**Data curation:** Eliz Kilich, Sara Dada, Mark R. Francis, John Tazare.

**Formal analysis:** Eliz Kilich, Sara Dada, John Tazare.

**Funding acquisition:** Heidi J. Larson.

**Methodology:** Eliz Kilich, Sara Dada, R. Matthew Chico.

**Software:** John Tazare.

**Supervision:** Pauline Paterson, Heidi J. Larson.

**Visualization:** Eliz Kilich, Sara Dada, John Tazare.

**Writing – original draft:** Eliz Kilich, Sara Dada, Mark R. Francis, John Tazare.

**Writing – review & editing:** Eliz Kilich, Sara Dada, Mark R. Francis, John Tazare, R. Matthew Chico, Pauline Paterson, Heidi J. Larson.

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
