## [Decision Letter · Decision Letter 0]

31 Mar 2020

PONE-D-20-05667

Factors that influence vaccination decision-making among pregnant women: a systematic review and meta-analysis

PLOS ONE

Dear Dr Kilich,

Thank you for submitting your manuscript to PLOS ONE. After careful consideration, we feel that it has merit but does not fully meet PLOS ONE’s publication criteria as it currently stands. Therefore, we invite you to submit a revised version of the manuscript that addresses the reviewers points raised below during the review process.

We would appreciate receiving your revised manuscript by May 15 2020 11:59PM. To enhance the reproducibility of your results, we recommend that if applicable you deposit your laboratory protocols in protocols.io, where a protocol can be assigned its own identifier (DOI) such that it can be cited independently in the future. For instructions see: http://journals.plos.org/plosone/s/submission-guidelines#loc-laboratory-protocols

We look forward to receiving your revised manuscript.

Kind regards,

Ray Borrow, Ph.D., FRCPath

Academic Editor

PLOS ONE

Journal Requirements:

1. lease ensure that your manuscript meets PLOS ONE's style requirements, including those for file naming. The PLOS ONE style templates can be found at http://www.plosone.org/attachments/PLOSOne_formatting_sample_main_body.pdf and http://www.plosone.org/attachments/PLOSOne_formatting_sample_title_authors_affiliations.pdf

2. We would strongly encourage you to update your search results to include any relevant studies that may have been published since November 2018.

"This research has been funded by a grant from GlaxoSmithKline to support research on maternal vaccination. The funders had no role in study design, data collection and analysis, decision to publish, or preparation of the manuscript."

We note that you received funding from a commercial source: GlaxoSmithKline

"HL’s research group (HL, PP, MF, EK, SD) has received funds from GlaxoSmithKline and Merck. This research has been funded by a grant from GlaxoSmithKline to support research on maternal vaccination. HL has served on the Merck Vaccines Strategic Advisory Board 2014-2016. None of the other authors have conflicts of interest to declare."

5. Please upload a copy of Supporting Information S1 Fig1 and S2 Fig2 which you refer to in your text on page 26.

6. We note that this manuscript is a systematic review or meta-analysis; our author guidelines therefore require that you use PRISMA guidance to help improve reporting quality of this type of study. Please upload copies of the completed PRISMA checklist as Supporting Information with a file name “PRISMA checklist”.

Reviewers' comments:

Reviewer's Responses to Questions

**Comments to the Author**

1. Is the manuscript technically sound, and do the data support the conclusions?

Reviewer #1: Yes

Reviewer #2: Yes

2. Has the statistical analysis been performed appropriately and rigorously? 

Reviewer #1: I Don't Know

Reviewer #2: Yes

3. Have the authors made all data underlying the findings in their manuscript fully available?

Reviewer #1: Yes

Reviewer #2: No

4. Is the manuscript presented in an intelligible fashion and written in standard English?

Reviewer #1: Yes

Reviewer #2: Yes

5. Review Comments to the Author

Reviewer #1: In general, looking more closely at what is understood about barriers to improving uptake of maternal vaccination is a very important undertaking. This study has provided a comprehensive examination of what has been published on decision making around vaccination for 2 routinely recommended vaccines, as well as the pandemic H1N1 vaccine. I think the overall methodology is in keeping for systematic review protocol and the authors adhered to rigorous processes for distilling the data into a useful summary. I did feel that there was a general lack of contextual information around the advancement of recommendations for influenza and pertussis-containing vaccines and how, at least in the US, we’ve observed shifts in the recommendation language in the study period being considered (1996-2018) for both influenza and Tdap vaccines, as well as the uptake of these vaccines. By including studies from such a broad study period, there have to be important differences based on the timing of the included publications. I don’t see any mention of this issue from the authors and feel that some discussion on this is warranted (more in Methods below). The challenge with including studies from around the world is that the recommendations and barriers really do vary by country. My second overarching issue has to do with study samples and methods for determining eligibility in the included studies. Some methodologic process should be used to assess how populations were sampled and surveyed (or interviewed) to address whether there are potential biases that could be introduced.

Intro:

Do the authors anticipate differences in decision-making factors for routine vs. outbreak vaccines? It is hard to imagine that the results would completely align and some mention of how the authors expect to see these interplay and diverge would be nice to introduce.

Methods:

• Inclusion of studies from all countries poses an interesting challenge. While it is nice to have perceptions from a more inclusive set of populations, the universality of messaging around vaccines may need to be different country by country. Presumably, different themes emerged by country/region and it would be nice to have a sense of how the authors treated this issue. The discussion section has a mention of it (line 479), but if there was any specific approach going in to account for this issue it would be nice to state outright in the methods.

• Did the authors consider examining HCP recommendation more finely (i.e. restricting to studies published after routine HCP recommendations were in place for the respective vaccines, in respective countries OR looking before/after H1N1, when vaccine uptake for seasonal influenza was noted to have had large increases)?

• No mention of how studies selected populations for survey collection—this is an important factor in data collection and potential biases in studies.

• Same as above re: qualitative studies. How were these women selected to take part in focus groups?

• In some places the authors refer to Tetanus vaccine, in other Pertussis vaccine. I’m not clear on whether there are recommendations for Td alone, in some countries, but it would be helpful if the authors explain this somewhat.

Results:

It is helpful to see an overall magnitude of effect for various associations with maternal vaccination uptake. It is also very useful to see that perceptions of disease risk or susceptibility, alone, do not necessarily increase likelihood of vaccination (or not as strongly as HCP recommendation) as this has direct implications for messaging around routine and novel vaccines.

Line 260: The authors state that “knowledge of specific pan flu vaccine side-effects”—can you please explain how “knowledge” is defined here?

Discussion:

Line 451: There are multiple cites that should be inserted here, referencing the Vaccine Safety Datalink work that has focused on safety of maternal vaccination.

in response to the author’s comment that HCP do not have ready access to clear language on safety of vaccine during pregnancy, this could be something to add in the conclusion paragraph, starting on line 488.

Reviewer #2: Abstract

Very clearly written

I don’t agree with the overall assertion that vaccine safety should be focused on at the expense of disease severity as your data don’t support this overall

Introduction

Line 88 - why is low coverage defined as < 70%. Suggest outlining what optimal coverage is. Also coverage varies widely across regions/countries – in most settings flu is much lower than pertussis and this should be stated

Page 5 – line 113 – which national contexts are you referring to? In every country?

Page 5 – line 132 – vaccine interventions “to increase uptake”

Methods

Line 6 – why were behavioural intervention studies excluded? Why were sociodemographic variables excluded?

Page 7 – second round of coding?

Page 7 – line 175 - ? separate doesn’t seem to fit here? Remove?

Results

Figure 1 – why were there over 9,000 duplicates removed – this seems exceptionally high?

30 studies were excluded from the meta-analysis - why was this - is this no reasons vaccines declined/accepted given?

35 articles had vaccination outcomes and 14 were only intention – how did you a count for this differential given that we know intention does not correlate highly with uptake?

Quant studies:

Page 16 – can you please outline how many studies had vaccine uptake vs intention – should this not be explored and reported separately? This is not clear that you actually used intention in reporting of your overall results NOT uptake

Line 238 – for the increased odds of vaccination, why have you just focused on HCP recommendation, previous vaccination and awareness of vaccination – belief that the disease is harmful for S flu has an OR of 3.7 and concern/awareness (not sure – not stated?) of hospitalisation for P flu is 2.91? Your focus is on disease susceptibility being non-sig when S flu disease susceptibility when pregnant is sig with OR 1.76 ? Why is belief in disease severity for S flu not highlighted as sig?? This is quite misleading and not consistent with other research we are aware of and have found in studies recently

- the focus of reduced odds is all on vaccine safety which is not quite correct

- Fig 2: should the title of this Table not be Factors influencing mat vacc FLU uptake? Where is the equivalent table for pertussis?

Qual studies

- page 18 – line 294 – suggest removing word oftentimes – replace with often

- here disease awareness, presumably severity/susceptibility, is highlighted as a finding but not reported in conclusion/abstract either; line 316 – this finding is again clearly stated here and again line 336

- more of a focus on vaccine safety

- line 341 – was this just for I study or all re fear of harm being main driver for rejection of vaccines compared fear of the disease?

Conclusion

Page 20 – line 351 – here it is stated that vaccine specific factors and previous vacc behaviours have a strong influence on decision making but that disease related perceptions only have a modest effect on UPTAKE – your data don’t support this and it is not clear whether you are talking about INTENTION or actual UPTAKE here?

- line 368 – again you refer to uptake – in the results you say mostly the analysis was with studies that reported intention – this is not clear?

- line 370 – again the focus on susceptibility and downplaying severity – the neg pandemic flu OR is highlighted here but not the belief that the disease is harmful for S flu which has an OR of 3.7 and concern/awareness (not sure – not stated?) of hospitalisation for P flu is 2.91 (both sig)? I am not sure it is accurate to pick certain results to support your assertions when your data say otherwise

- line 377 – your findings actually don’t support that messaging in vaccine campaigns should not focus on disease threat or hospitalisation for seasonal flu

- page 22 – line 397 – I don’t think I agree with this assertion for the above reasons

- page 24 – line 441 – again I disagree here – rather than making them fear the disease, awareness and explanation of disease risk is important to weight the perceived vaccine risks

- page 25 – line 472 – limitations - it is very unclear which studies you had no intention/uptake data, just intention or just uptake – if this is the case how can all your findings be presented in terms of vaccine uptake??

- page 25 – line 495 – I agree that communication should focus on safety but that disease threat needs to be addressed as well – it is not one or the other and your data don’t actually support the disease threat for S flu was not significant in predicting vaccine “uptake”

My overall concern is that your interpretation of results data don’t clearly allow the reader to determine whether these factors are related to vaccine uptake or intention or neither and that suggestion to preferably focus on vaccine safety seems to be at the expense of addressing disease severity which is dangerous. I think this needs to be more balanced. Also this data seems to only be presented in Fig 2 for flu and not for pertussis? For overall clarity I think all these issues should be addressed

6. PLOS authors have the option to publish the peer review history of their article (what does this mean?). If published, this will include your full peer review and any attached files.

Reviewer #1: No

Reviewer #2: No

---

## [Author Response · Author response to Decision Letter 0]

28 Apr 2020

Comment 1: Please ensure that your manuscript meets PLOS ONE's style requirements, including those for file naming. 

Response: The manuscript has been formatted to meet the style requirements. The abstract has been addended to meet the 300-word limit. 

Comment 2: We would strongly encourage you to update your search results to include any relevant studies that may have been published since November 2018.

Response: Thank you for this suggestion. We agree in the value of updating our systematic review through April 2020. However, conducting new searches across 10 databases poses unique challenges during the current COVID-19 pandemic, especially as we do not have key hardware and software readily available. Given the COVID-19 outbreak, we believe the benefit of timely dissemination of our findings outweighs the added value from an updated search. Our quantitative data – yielded from a review of over 17,000 records – is unique in scope and unlikely to change materially given the comprehensive inclusion of studies over a long period of time (studies included data collected in August 1996 and published through November 2018). We hope the editor might agree, particularly under the present circumstances. 

In addition, and within the current context of the COVID-19 and vaccine development, we believe it is important to disseminate the findings from this study in a timely manner as vaccination programmes plan for the rollout of novel vaccines. The lessons from this study can be directly guide and enhance communications around a maternal COVID-19 vaccine. We highlight the findings that suggest the influence of a healthcare provider recommendation for vaccination during a pandemic is weaker compared to routine antenatal vaccinations. We also highlight the key finding that if there is doubt in the safety of the vaccination, there may be reduced uptake irrespective of the belief in disease risk. We present data to suggest that fear in disease risk could amplify this response to remain unvaccinated and so careful communication strategies should be tested and applied as part of the COVID-19 response. From a research strategy perspective, this study also highlights the challenges that arise in synthesizing very diverse datasets. This finding suggests and calls for unification in research methodology (including unified survey design) to harmonise data collection. The importance of this cannot be overstated as many research studies will investigate the acceptability of a COVID-19 vaccine later this year.

We hope the editor might agree, particularly under the present circumstances.

Comment 3: Please provide an amended Competing Interests Statement that explicitly states this commercial funder, along with any other relevant declarations relating to employment, consultancy, patents, products in development, marketed products, etc. Within this Competing Interests Statement, please confirm that this does not alter your adherence to all PLOS ONE policies on sharing data and materials by including the following statement: "This does not alter our adherence to PLOS ONE policies on sharing data and materials.” (as detailed online in our guide for authors) 

Response: We have added this statement to the manuscript (Page 21, Lines 588-593). 

Comment 4: Please upload a copy of Supporting Information S1 Fig1 and S2 Fig2 which you refer to in your text on page 26.

Response: We uploaded these files on initial submission. We have re-uploaded these files as suggested. If you are still unable to download them please contact us again.

Comment 5: We note that this manuscript is a systematic review or meta-analysis; our author guidelines therefore require that you use PRISMA guidance to help improve reporting quality of this type of study. Please upload copies of the completed PRISMA checklist as Supporting Information with a file name “PRISMA checklist”.

Response: We originally included a PRISMA checklist in the Appendix on p16 Supplementary File Number 4 S3. To ease locating the PRISMA checklist for readers, it now is a stand-alone file in supporting information S3 PRISMA Checklist. 

Comment 6: Do the authors anticipate differences in decision-making factors for routine vs. outbreak vaccines? It is hard to imagine that the results would completely align and some mention of how the authors expect to see these interplay and diverge would be nice to introduce.

Response: We hypothesized that the decision-making factors would diverge in the case of an outbreak. We agree that it is hard to imagine that they would align. To date a comparison of vaccines used in the outbreak setting versus routinely in pregnant women is not well-covered within the literature. We believe the context surrounding the decision is altered, which has subsequent effects on the perception of the disease and the vaccine. The perception of disease risk may be heightened by socio-political factors (e.g. news coverage/social media, fear and mortality risk) yet the perceptions of vaccine risk may also accordingly be heightened if it is a novel vaccine. We included the following sentence on Page 3, line 90-91: 

“The concern of disease risk may be amplified during an outbreak, but concerns about using a novel vaccine may also be enhanced.”

Comment 7: Line 88 – a) why is low coverage defined as < 70%. Suggest outlining what optimal coverage is. b) Also, coverage varies widely across regions/countries – in most settings flu is much lower than pertussis and this should be stated.

Response: a) Having an optimal coverage defined would be ideal. Neither the WHO, Public Health England or the CDC provide a consensus statement on the optimal national maternal coverage for antenatal vaccination. This makes defining low coverage challenging. For clarity we have changed the word on Page 3, Line 76 : “low” to “suboptimal” given optimal would be close to 100% without an additional reference. We used 70% here to reflect a summary of the coverage rates of maternal influenza and pertussis vaccines in the UK and the USA. Public Health England data suggests coverage is approximately 70% for pertussis vaccination. There is no meta-analysis in the literature on the coverage of maternal influenza and pertussis vaccines globally. Most countries do not have data reporting national coverage hence we restricted extrapolating our data on coverage beyond the US/ UK who provide national data. We highlight this on Page 3, Line 77-79:

“Suboptimal maternal vaccination coverage (estimated between 0-70%) of seasonal influenza and pertussis vaccines globally represents a missed opportunity to improve maternal and neonatal health [3-5].”

We provide data on vaccine coverage of participants within the individual studies included in the systematic review (Table 1). It must be noted that in the studies included in the systematic review the samples included were not nationally representative with the majority of participants derived from convenience and maximal samples. 

b) We discuss the coverage levels deemed to be nationally representative across the USA (distinguishing between influenza and pertussis) on Page 4, Lines 111-115 we state: 

“In 2018, the Centers for Disease Control and Prevention (CDC) found that 79.3% of pregnant participants received a recommendation or an offer for Tdap vaccine, but 45.6% of them chose to remain unvaccinated [4]. For seasonal influenza, fewer women chose to vaccinate when recommended to do so; 81.1% received a recommendation or an offer yet 50.9% of pregnant women surveyed remained unvaccinated [4]. Understanding why women remain unvaccinated despite an HCP recommendation is key.” 

This reflects that influenza coverage is lower than Tdap albeit alongside an additional point. We agree it will be important to be more explicit here. To distinguish the points, we have added a sentence to follow on from this statement to indicate that this also reflects the global coverage. Page 4, Lines 116-118: 

“We also sought to discriminate factors that influence specific vaccines since seasonal influenza vaccination coverage is lower than other routine vaccines (Tdap, tetanus) during pregnancy.” 

Comment 8: Page 5 – line 113 of original manuscript – which national contexts are you referring to? In every country?

Response: This is referring to data by individual countries– for most countries (particularly low and middle income) the data on vaccine coverage is not available at a national level. Estimates of vaccine coverage may be extrapolated from studies which look at coverage at a state or regional level (see Table 1 of manuscript). However, the data is somewhat limited and often not national representative. Some countries may have this data on a national level, but it is not a metric available publicly. The wording of this sentence has been adjusted for clarity on Page 3, Lines 106-107:

 “In general, there is limited data on maternal vaccination uptake and records of HCP recommendations at a national level.”

Comment 9: Page 5 – line 132 – vaccine interventions “to increase uptake”

Response: This wording on Page 4, Line 129 has been changed.

Comment 10: Inclusion of studies from all countries poses an interesting challenge. While it is nice to have perceptions from a more inclusive set of populations, the universality of messaging around vaccines may need to be different country by country. Presumably, different themes emerged by country/region and it would be nice to have a sense of how the authors treated this issue. The discussion section has a mention of it (line 479), but if there was any specific approach going in to account for this issue it would be nice to state outright in the methods.

Response: Prior to obtaining the results we planned to conduct a sensitivity analysis by Gross Domestic Product (GDP) of countries included in the systematic review. Because there were few numbers of studies included into each meta-analysis, it was not possible to conduct sensitivity analyses to take this into account. 

Before we undertook the study, in our protocol we had hoped that the survey questions across studies would be similar/ if not exactly identical. However, the results indicated that the wording across studies included in a proposed “theme” were frequently different. This meant that the authors had to generate several rounds of coding until the questions were suitably summated, excluding studies that could not be summated (Results of those studies excluded in Appendix). 

This limited the number of studies that could be reliably summated. For example, the theme “disease severity” the authors had to divide this theme into subcodes e.g. into “disease can kill you” “disease can cause hospitalization” since these were the exact phrasings used in surveys. Therefore, rather than 1 meta-analysis on disease severity we generated two separate meta-analyses: the first being a belief in disease hospitalization and a second being a belief in disease causing death. This reduced the number of studies that could be included in each meta-analysis. Since the number of studies included in each meta-analysis was frequently <5, a sensitivity analysis of GDP/ by location was inappropriate and uninterpretable on our attempts. 

However, the qualitative study synthesis does make note of some of the nuances of the findings across the different country settings. This is a limitation of the study which we highlight on Page 20, Lines 535-537:

“Our findings were largely consistent across countries; however, we recognize that the majority of data come from high-income settings where national vaccine policies exist and, therefore, the generalisability of these findings may be limited.”

We have added a sentence on Page 20, Lines 537-539: 

“We were unable to conduct a sensitivity analysis to stratify results by Gross Domestic Product of each country given that there were too few studies included in each meta-analysis.” 

We have also included this sentence into the methodology on Page 5, Lines 183-185: 

“We wished to conduct a sensitivity analysis assessing the robustness of results by Gross Domestic Product of countries included to assess the influence of geographical context.”

Comment 11: Did the authors consider examining HCP recommendation more finely (i.e. restricting to studies published after routine HCP recommendations were in place for the respective vaccines, in respective countries OR looking before/after H1N1, when vaccine uptake for seasonal influenza was noted to have had large increases)?

Response: This is a very interesting question. Our statistician has performed an analysis on your suggestion. Including studies which were conducted after 2009-2010 period, the pooled OR for the effect of HCW recommendation on seasonal flu uptake is 11.1 (95% CI 6.1-20.0 I2 = 92.6) whereas prior to the pandemic the increased odds of receiving the seasonal flu vaccine by HCW recommendation was 32.83 (95% CI 11.5-93.9 I2=0). It is difficult to compare because there are only two studies prior to 2009 on seasonal flu that were suitable for meta-analysis. We have not included the statistic in our write-up given the data cannot facilitate the comparison and may be misleading. 

Because we aimed to perform a global analysis rather than a country-specific analysis, the importance of time is difficult to assess as countries introduced routine vaccination of seasonal influenza and pertussis vaccination in different years. For instance, looking specifically at pertussis, in the UK following the 2012 outbreak, pertussis vaccine was recommended routinely by HCW. However, this was instated by Israel in 2015, Australia in 2013, the USA in 2011. This is, nonetheless, a critical question and we hope future studies could be designed to answer it (particularly during and after the COVID-19 pandemic). 

Given this reviewer has raised a very important point about context including timing of recommendations, we have included the following in the Introduction to provide readers a better context for interpreting the data on Page 3, Lines 63-76: 

“The World Health Organisation (WHO) recommends the inactivated influenza, tetanus-toxoid-containing vaccine (TTCV), and combined tetanus, diphtheria, and acellular pertussis (Tdap) vaccines for pregnant women in settings where the disease burden is known [2]. Historically, maternal tetanus vaccination was limited to areas of significant transmission. In areas where there is ongoing maternal to neonatal transmission of tetanus, two doses of TTCV (preferably Tetanus-diphtheria) are recommended in pregnancy in addition to Tdap or DTaP (for pertussis) and seasonal influenza vaccines.[2] Pertussis vaccination was limited to childhood however the resurgence of pertussis during outbreaks that disproportionately affected younger infants, led to national policy changes between 2011 and 2015, in countries such as the United Kingdom and the United States, that introduced routine maternal pertussis vaccination.[2-3] Similarly, the widespread influenza immunisation programs during the 2009 H1N1 pandemic resulted in public health bodies particularly in Europe, the United States and Australia introducing guidance to implement recommendations for routine annual seasonal influenza vaccination during the subsequent decade.”

Comment 12: No mention of how studies selected populations for survey collection—this is an important factor in data collection and potential biases in studies.

Response: We have now included the sampling method as a column in the summary table (Table 1) on Pages 7-11. We took into consideration the potential biases posed by survey sampling within the JBI critical appraisal (question category: sample inclusion). Other factors that were considered in the JBI critical appraisal of all included studies were around: how exposures were measured, identifying and controlling for confounding variables, using appropriate statistical analysis methods. Therefore, the biases of the papers 

The JBI critical appraisal tool was used to provide an assessment of the quality of studies included and therefore the risk of bias in these studies. This was then used to perform a sensitivity analysis (of high quality/low quality), which is presented in the Appendix. However, given the low number of papers in each meta-analysis, it is difficult to interpret the relevance of this. We have included a sentence within our limitations section on Page 20, Lines 543-547: 

“A number of studies recruited participants using purposive or convenience sampling (Table 1). Thus, we applied the Joanna Briggs Institute (JBI) Critical Appraisal Tools to assess potential biases among individual studies. Based on our results, we then performed a pre-defined sensitivity analysis (of high quality/low quality) presented in the Appendices (Appendix 22, Appendix 24 p90, p92). We were unable to detect an influenced study quality on our pooled analyses.”

Comment 13: Same as above re: qualitative studies. How were these women selected to take part in focus groups?

Response: We have included the sampling method as a column in the summary table (Table 1) on Pages 7-11.

Comment 14: In some places the authors refer to Tetanus vaccine, in other Pertussis vaccine. I’m not clear on whether there are recommendations for Td alone, in some countries, but it would be helpful if the authors explain this somewhat.

Response: For many low-income countries where there is still maternal to neonatal transmission of tetanus - maternal one-off dosing of the tetanus, diphtheria, & acellular pertussis (Tdap) vaccine is not the routine standard of care as multiple doses of the tetanus toxoid are required to provide sufficient protection (insufficient with one dose of Tdap). Tdap is not recommended in multiple doses during a single pregnancy. For these countries they do not have data on their pertussis burden to warrant pertussis vaccination (e.g. Tdap or DTaP/IPV). 

Maternal tetanus, diphtheria, & acellular pertussis (Tdap) vaccine (US/Aus) or Diphtheria, tetanus, acellular Pertussis/ Inactivated polio vaccine (DTaP/IPV) (UK) regimens are designed to protect against pertussis with some additional booster protection against Tetanus. For high income countries with sufficient childhood and adult booster programs for tetanus additional maternal tetanus toxoid vaccine (TT) or tetanus & diphtheria vaccine (Td) is not required and not the standard of care.

However, in countries where the health priority is to protect against tetanus (because maternal-neonatal transmission is ongoing) they use a specific programme of the monovalent tetanus toxoid vaccine (TT) or tetanus & diphtheria (Td) vaccination as advised by WHO. The vaccine regimen is more intensive to ensure protection and for women who have not received their correct childhood doses or booster they will require. 

The 2017 WHO Position paper on Tetanus (referenced in the manuscript) states that: “In countries where MNT remains a public health problem, pregnant women for whom reliable information on previous tetanus vaccinations is not available should receive at least 2 doses of TTCV, preferably Td, with an interval of at least 4 weeks between doses and the second dose at least 2 weeks before the birth. To ensure protection for a minimum of 5 years, a third dose should be given at least 6 months later. A fourth and fifth dose should be given at intervals of at least 1 year, or in subsequent pregnancies, in order to ensure lifelong protection. Pregnant women who have received only 3 doses of TTCV during childhood without booster doses should receive 2 doses of TTCV at the earliest opportunity during pregnancy with a minimal interval of 4 weeks between doses and the second dose at least 2 weeks before giving birth. Although 1 booster dose should result in a rapid increase in antibody, the level of tetanus-specific antibodies in women who received only a 3-dose primary series during infancy is similar to that of unimmunized individuals 15 years post-immunization. Therefore, 2 doses are recommended in order to ensure a total of 5 doses before delivery.” 

Quick access to WHO Position Paper https://apps.who.int/iris/bitstream/handle/10665/254582/WER9206.pdf;jsessionid=54F2205732E1EC20F1497B50C6813D5E?sequence=1

Page 3, Lines 67-69 now state: 

“In areas where there is ongoing maternal to neonatal transmission of tetanus two doses of TTCV (preferably Td) are recommended in pregnancy alongside Tdap (for pertussis) and seasonal influenza vaccines.” 

Comment 15: Line 6 – Why were behavioural intervention studies excluded? Why were sociodemographic variables excluded?

Response: Behavioural intervention studies were excluded because the population were deemed to be different to the baseline population in standard survey studies. We hypothesized that being recruited into a study may make them to be prone to particular beliefs or attitudes (increased willingness to adopt healthcare views). Additionally, data reported in behavioural intervention studies were more likely reflect changes in perceptions rather than baseline characteristics. 

Sociodemographic variables were excluded as they were beyond the scope of this review. We wished to understand potentially modifiable factors related to vaccine uptake or intention. 

Comment 16: Page 7 – second round of coding?

Response: Second round of coding refers to coding into subcategories based on common themes within the broad 1st round of codes. See appendix 11, Pages 40-43. Page 5, Lines 165-170 have been rearranged for clarity so that the new passage explaining the process of coding all data:

“The quantitative studies were independently assessed (EK, MF) for inclusion in the meta-analysis based on first cycle broad codes to capture data that could be synthesized (appendix p38-40). Qualitative data underwent a second round of coding to identify specific patterns within the broad themes (Inter-rater reliability kappa score 0.88). A third round of coding (subdividing the first cycle codes) was conducted to ensure that only data that was directly comparable were included in each meta-analysis (appendix p41).”

Comment 17: Page 7 – line 175 - ? separate doesn’t seem to fit here? Remove?

Response: This typo has been corrected by removal

Comment 18: Line 260: The authors state that “knowledge of specific pan flu vaccine side-effects”—can you please explain how “knowledge” is defined here?

Response: Knowledge was defined as a known adverse reaction as outlined by the drug company label insert, e.g. fever. We have added to Page 14-15 Line 271-274 

“having knowledge of specific pandemic influenza vaccine side-effect (defined as a awareness of a known adverse reaction as outlined by the drug company label insert, e.g. fever.) (OR 0.27, 95% CI 0.21-0.34, 2 studies, 1,325 women) [41, 53] was associated with three-times lower odds of vaccination.”

Comment 19: Figure 1 – why were there over 9,000 duplicates removed – this seems exceptionally high?

Response: Our systematic review spanned 10 databases curated by multiple institutions with indexing that generates extensive overlap. We chose to search widely to ensure the inclusion of the greatest possible of publications, anticipating high numbers of duplicates in the process. 

Comment 20: 30 studies were excluded from the meta-analysis - why was this - is this no reasons vaccines declined/accepted given?

Response: All reasons for exclusion from the meta-analysis are included in Appendix 14 (Pages 44-50) and the reasons for vaccines being declined/accepted in these papers are presented in Appendix 16. These reasons included: (1) the exposure was not assessed, (2) the exposure measurement was not discrete, (3) insufficient data, and (4) the exposure was not assessed in relation to the outcome. For the papers that were excluded from the analysis the reasons for declining and accepting are copied below from Appendix 16:

Comment 21: 35 articles had vaccination outcomes and 14 were only intention – how did you a count for this differential given that we know intention does not correlate highly with uptake?

We have now included the sampling method for individual studies as a column in the summary table (Table 1) on Pages 7-11. We took into consideration the potential biases in studies using the Joanna Briggs Institute (JBI) Critical Appraisal Tools. JBI appraisal involves assessing potential biases that may result from how exposures were measured, potential confounding, and use of appropriate statistical methods. Based on our results, we performed sensitivity analysis (of high quality/low quality) presented in the Appendix. We were unable to detect an influenced study quality on our pooled analyses which we note in our limitations section. Page 20, Lines 543-547:

“A number of studies recruited participants using purposive or convenience sampling (Table 1). Thus, we applied the Joanna Briggs Institute (JBI) Critical Appraisal Tools to assess potential biases among individual studies. Based on our results, we then performed a pre-defined sensitivity analysis (of high quality/low quality) presented in the Appendices (Appendix 22, Appendix 24 p90, p92). We were unable to detect an influenced study quality on our pooled analyses.”

Page 14, Line 234-236: 

“The majority of studies with quantitative results for the pertussis vaccine had investigated intention to be vaccinated rather than actual vaccination status.”

Page 15, Lines 299-305: 

“Although based on a small number of studies, there was incomplete agreement between the primary and secondary analyses (intention to be vaccinated meta-analysis results are provided in appendix on pages 91-92). Sensitivity analyses including studies with Joanna Briggs Institute scores >10 were conducted for both vaccination status and intention to be vaccinated outcomes (appendix: page 90 and 92). Whilst results were generally consistent, differences were difficult to interpret due to the low number of higher-quality studies.”

Comment 22a): Quant studies: Page 16 – can you please outline how many studies had vaccine uptake vs intention – should this not be explored and reported separately? This is not clear that you actually used intention in reporting of your overall results NOT uptake

Response: We did analyse these separately and report them separately. Please see PRIMSA diagram (Fig1 S1) (14 studies Intention only and 35 studies vaccination outcome). An analysis of the results distinguishing intention to vaccinate AND vaccination outcome can be found in the Appendix and Manuscript (respectively). Appendix 18-Appendix 24 explores all the summary statistics and summary tables for a) primary outcome (vaccination uptake) and b) secondary outcome (intention to vaccinate). Appendix 23 - Table – Summary ORs from secondary analysis: meta-analyses investigating association between beliefs/experiences and intention to vaccinate. For clarity we have included the following statement in our results Page 14, Line 228-234:

“For our primary analysis we conducted 33 meta-analyses which assessed the relationship between a specific belief or behaviour and maternal vaccination status (338-14,099 participants (average 2,955) included in each meta-analysis)) (appendix p55-89 for individual meta-analyses and summary table) (Fig 2). For our secondary analysis we conducted 15 meta-analyses assessing the relationship between a specific belief or behaviour and maternal vaccination intentions, rather than prior behaviour (531-2,215 participants (average 1,344) included in each meta-analysis)) (appendix p91).”

b) Line 238 – for the increased odds of vaccination, why have you just focused on HCP recommendation, previous vaccination and awareness of vaccination – belief that the disease is harmful for S flu has an OR of 3.7 and concern/awareness (not sure – not stated?) of hospitalisation for P flu is 2.91? Your focus is on disease susceptibility being non-sig when S flu disease susceptibility when pregnant is sig with OR 1.76? Why is belief in disease severity for S flu not highlighted as sig?? This is quite misleading and not consistent with other research we are aware of and have found in studies recently

Response: We agree with this comment regarding S flu, although the reviewer may have not seen our related statement in the original manuscript on Page 15, Lines 293-295, which state: 

“However, pregnant women who believed seasonal influenza could be harmful to their pregnancy or baby had four-times greater odds of being vaccinated than those who did not believe seasonal influenza could affect their pregnancy or baby (OR 3.70, 95% CI 1.37-9.94, 3 studies, 1,748 women) [22, 30, 47].”

We have added an additional comment to indicate this result is not in line with the rest of the “inconclusive data.” Page 15, Lines 289-296

“There was inconclusive evidence to support a similar association between perceptions of the severity of pandemic (OR 2.04, 95% CI 0.98-4.26, 4 studies, 5,948 women) [41, 43, 52, 53] or seasonal influenza (OR 1.56, 95% CI 0.88-2.76, 4 studies, 2,671 women) [23-25, 38] with vaccination status. However, pregnant women who believed seasonal influenza could be harmful to their pregnancy or baby had four-times greater odds of being vaccinated than those who did not believe seasonal influenza could affect their pregnancy or baby (OR 3.70, 95% CI 1.37-9.94, 3 studies, 1,748 women) [22, 30, 47].”

With regards to susceptibility we state on Page 15, Lines 286-290: 

“However, pregnant women who felt they were susceptible to seasonal influenza had almost two-times higher odds of vaccination than those who did not feel susceptible to contracting seasonal influenza (OR 1.76, 95% CI 1.26-2.47, 5 studies, 4,763 women) [23-25, 39, 48].”

The relationship between seasonal flu and disease severity was not outlined as being significant because it crossed the confidence interval, Page 15, Lines 292: 

“seasonal influenza (OR 1.56, 95% CI 0.88-2.76, 4 studies, 2,671 women) [23-25, 38] with vaccination status.”

The reviewer has raised some important points here. The absence of statistical power is not the same as a lack of importance; our data suggests that the role of the belief in disease severity and susceptibility is inconclusive, not unimportant. We have adapted our discussion accordingly (Comment 23d, Comment 28, Comment 31). 

c) The focus of reduced odds is all on vaccine safety which is not quite correct

Response: We are not entirely sure what is meant by this comment, although we are happy to provide additional clarification if something is not clear. We have focused on the reduced odds related to vaccine harm and side effects because that what is reported in the literature. Please see Figure 2 Forest Plot. Supplementary File 2 (S2 Fig 2). We report “reduced odds” and “increased odds” based on the forest plot which used the data from the included literature.

d) Fig 2: should the title of this Table not be Factors influencing mat vacc FLU uptake? Where is the equivalent table for pertussis?

Response: The Table was intended to record the data for all our primary data e.g. “vaccination uptake” (rather than our secondary outcome “vaccination intention”). We would like to keep the Table name the same, although will defer to this decision to the editor. Unfortunately, no study which analysed pertussis looked at our primary outcome (vaccination uptake) and so we could not report it here. However, the table should still reflect our aim which is our primary outcome for all vaccines (seasonal flu, pertussis, pandemic flu, tetanus). Sadly, the data was not available for pertussis or tetanus. We make note of this in the Discussion section, Pages 18-19, Lines 528-530: 

“This is important since the majority of data available for pertussis vaccination in pregnancy focused on beliefs associated with vaccination intentions only [80, 87, 94, 95].”

Comment 23: Qual studies a) page 18 – line 294 – suggest removing word oftentimes – replace with often

Response: We have adjusted this phrasing (Page 15, Line 310). 

b) Here disease awareness, presumably severity/susceptibility, is highlighted as a finding but not reported in conclusion/abstract either; line 316 – this finding is again clearly stated here and again line 336.

Response: Our qualitative analysis noted vaccine and/or disease awareness and fear as factors in vaccine decision-making, e.g. on Page 15, Lines 317-319: 

“Almost all qualitative studies indicated that being aware of maternal vaccination and/or the respective disease, regardless of information source, was key to receiving the vaccine but rarely sufficient (17 studies) [55, 57-72].” 

And Page 16, Line 354-357: 

“Pregnant women feared the unknown [67, 69-71], the disease (particularly for pandemic influenza) [61, 67, 69, 72], vaccine harm.” 

However, when we combine this with the rest of the qualitative analysis and quantitative analysis we find that the effect of this may be muted by the fear of vaccine harm. Therefore, our conclusion reflects a summation of this. We suggest that the belief in disease severity or susceptibility is important, however, when combined with other factors it may cause a counter-response to vaccine uptake. Page 19, Lines 478-480: 

“This was most notably captured by Meharry et al. as, “…fear if I do (vaccinate), fear if I don’t (vaccinate), and do nothing when I fear both” [72].”

If it is helpful for future readers, we can include a box insert or figure with the key definitions of the factors used throughout the paper. We defer this decision to the editor. 

a) More of a focus on vaccine safety

Response: This was one of our key findings.

b) Line 341 – was this just for I study or all re: fear of harm being main driver for rejection of vaccines compared fear of the disease?

Response: We noted that within our qualitative studies fear of harm of the vaccine if present along with fear of disease often resulted in inaction. Of the 19 qualitative studies that mention ‘risk of vaccine harm,’ 12 studies (Barroso Pereira 2013, Bettinger 2016, Cassady, Kharbanda 2011, Lohiniva, Lohm, Maisa 2018, Marsh 2014, McQuaid 2016, Meharry 2013, O’Shea 2018, Richun 2018, Yuen 2016) described cases where a fear or perception of the vaccine causing harm were a primary reason they rejected the vaccine, in these cases they explicitly stated their concern for potential effects of the illness was outweighed by their concern for potential effects of the vaccine. 

Uncertainty about the vaccine’s safety was described as a key obstacle to inﬂuenza vaccination (Yuen 2016). For example: Bettinger et al stated “For most, the unknown risks from the vaccine did not outweigh the beneﬁts of vaccination (posed by the risk of disease)”. Some were concerned that adverse effects of vaccination might be identiﬁed years in the future. Lynch et al provides further weight to this by stating “the seasonal flu vaccine was seen as safe and tested compared with the 2009 H1N1 vaccine which was seen as new and potentially unsafe (irrespective of the pandemic flu being viewed as more dangerous).” Additionally, the meta-analysis data also suggests awareness of disease severity and susceptibility is inconclusive. 

We have added the additional references. Page 16, Lines 358-361

“The fear of perceived vaccine harms (including the ideas of unknown risks for novel vaccines) were used to explain the rejection of maternal vaccination despite a connected fear of the disease it was aimed to protect against [58, 63-67, 69-72, 74].”

Comment 24: Line 451: There are multiple citations that should be inserted here, referencing the Vaccine Safety Data - link work that has focused on safety of maternal vaccination.

Response: Ten new references to Vaccine Safety Data have been included on Page X, Line Y.

92. Sukumaran L, McCarthy N, Kharbanda E, Vazquez-Benitez G, Lipkind H, Jackson L et al. Infant Hospitalizations and Mortality After Maternal Vaccination. Pediatrics. 2018;141(3):e20173310.

93. DeSilva M, Vazquez-Benitez G, Nordin J, Lipkind H, Klein N, Cheetham T et al. Maternal Tdap vaccination and risk of infant morbidity. Vaccine. 2017;35(29):3655-3660.

94. McMillan M, Clarke M, Parrella A, Fell D, Amirthalingam G, Marshall H. Safety of Tetanus, Diphtheria, and Pertussis Vaccination During Pregnancy. Obstetrics & Gynecology. 2017;129(3):560-573.

95. DeSilva M, Vazquez-Benitez G, Nordin J, Lipkind H, Romitti P, DeStefano F et al. Tdap Vaccination During Pregnancy and Microcephaly and Other Structural Birth Defects in Offspring. JAMA. 2016;316(17):1823.

96. Sukumaran L, McCarthy N, Kharbanda E, Weintraub E, Vazquez-Benitez G, McNeil M et al. Safety of Tetanus Toxoid, Reduced Diphtheria Toxoid, and Acellular Pertussis and Influenza Vaccinations in Pregnancy. Obstetrics & Gynecology. 2015;126(5):1069-1074.

97. McMillan M, Porritt K, Kralik D, Costi L, Marshall H. Influenza vaccination during pregnancy: A systematic review of fetal death, spontaneous abortion, and congenital malformation safety outcomes. Vaccine. 2015;33(18):2108-2117.

98. Moro P, Baumblatt J, Lewis P, Cragan J, Tepper N, Cano M. Surveillance of Adverse Events After Seasonal Influenza Vaccination in Pregnant Women and Their Infants in the Vaccine Adverse Event Reporting System, July 2010–May 2016. Drug Safety. 2016;40(2):145-152.

99. Moro P, Broder K, Zheteyeva Y, Walton K, Rohan P, Sutherland A et al. Adverse events in pregnant women following administration of trivalent inactivated influenza vaccine and live attenuated influenza vaccine in the Vaccine Adverse Event Reporting System, 1990-2009. American Journal of Obstetrics and Gynecology. 2011;204(2):146.e1-146.e7.

100. Naleway AL, Irving SA, Henninger ML, Li DK, Shifflett P, Ball S, Williams JL, Cragan J, Gee J, Thompson MG. Safety of influenza vaccination during pregnancy: a review of subsequent maternal obstetric events and findings from two recent cohort studies. Vaccine. 2014 May 30;32(26):3122-7.

101. Kharbanda EO, Vazquez-Benitez G, Lipkind H, Naleway A, Lee G, Nordin JD, Vaccine Safety Datalink Team. Inactivated influenza vaccine during pregnancy and risks for adverse obstetric events. Obstetrics & Gynecology. 2013 Sep 1;122(3):659-67.

Comment 25: In response to the author’s comment that HCP do not have ready access to clear language on safety of vaccine during pregnancy, this could be something to add in the conclusion paragraph, starting on line 488. 

Response: We have included this on Page 19, Lines 499-500: 

“We recommend that HCP are given ready access to clear language on the safety of vaccine during pregnancy”.

Comment 26: Page 20 – line 351 – here it is stated that vaccine specific factors and previous vaccine behaviours have a strong influence on decision making but that disease related perceptions only have a modest effect on UPTAKE – your data don’t support this and it is not clear whether you are talking about INTENTION or actual UPTAKE here?

Response: We understand this was not as clear as it could have been. To improve clarity, we have changed the word “decision-making” to “uptake”. Please see above regarding Comment 21 and Comment 22 as to that reflecting our primary analysis (e.g. vaccine uptake). Our appendix shows the results for our secondary analysis. 

With regard to the Reviewers comment that disease-related perceptions have a modest effect – this is based on the sizes of the odds ratio. For P Flu disease severity and susceptibility range from 1.11 (0.56-2.19) – 2.91 (2.02 -4.18) see Table below (Also found in Appendix 22 p 88). For seasonal Flu this ranges from 0.57 (0.22-1.45) to 3.70 (1.37- 9.94). Please note awareness and information is defined as having basic awareness of the disease and/or vaccine policy recommendations not a relationship to disease risk per se.

Whereas for vaccine-specific factors – the size of odds ratios was larger (See Forest Plot Figure 2 S2). For Pandemic Flu this is 0.19 (0.09-0.40) for ‘harm to baby’ (equivalent to an odds ratio of 5.3 in the reverse direction) and OR 0.16 (0.09-0.19) for ‘general harm’ (equivalent to a magnitude of 6.25)

Comment 27: line 368 – again you refer to uptake – in the results you say mostly the analysis was with studies that reported intention – this is not clear?

Response: We did analyse and report vaccine uptake and intention separately. Our manuscript results section is about vaccine uptake (primary outcome), not vaccine intention (secondary outcome). The latter is found in the Appendix. 

We used vaccine status (vaccinated, unvaccinated) as a surrogate for uptake. Please see Appendix for a full break down of results. 

Line 368 of original manuscript is consistent with our primary results. Our primary analysis is aimed at evaluating how factors influenced e.g. “vaccination uptake” (rather than our secondary outcome “vaccination intention”). For instance, earlier in the manuscript when we refer to results Page 14, Line 236-238

“..women who had received an HCP recommendation had 12-times higher odds of accepting seasonal 229 influenza vaccination (OR 12.02, 95% CI 6.80-21.23, 21 studies, 14,099 women) [20-40] and 10-230 times greater odds of accepting pertussis vaccine (OR 10.33, 95% CI 5.49-19.43, 2 studies, 637 231 women) [26, 28] compared to those who had not received recommendations.” 

This refers to uptake, not intention. Please see Summary Appendix tables 18 and 23. Unfortunately, no study which analysed pertussis also looked at our primary outcome (vaccination uptake) and so there was nothing to report. To clarify this, we have added a note in the Discussion section Pages 20, Lines 528-530: 

“This is important since the majority of data available for pertussis vaccination in pregnancy focused on beliefs associated with vaccination intentions only [80, 87, 94, 95].”

Comment 28: line 370 – again the focus on susceptibility and downplaying severity – the neg pandemic flu OR is highlighted here but not the belief that the disease is harmful for S flu which has an OR of 3.7 and concern/awareness (not sure – not stated?) of hospitalisation for P flu is 2.91 (both sig)? I am not sure it is accurate to pick certain results to support your assertions when your data say otherwise

Response: With a desire to be concise, we did not originally include all the data in the discussion. However, there is value in publishing all data on disease severity and susceptibility which we have now for S flu and P flu on Page 17, Lines 397-407 of the edited manuscript. We state that future studies should better characterize the relationship between disease severity and susceptibility when other beliefs such as in vaccine harm, or lack of recommendation are present: 

“There was some evidence to support an association between perceptions of the severity of pandemic influenza and pregnant women’s vaccination status (OR 2.04, 95% CI 0.98-4.26) with the belief pandemic influenza can result in hospitalisation increasing vaccine uptake 3-fold (OR 2.91, 95% CI 2.02-4.18). We would recommend additional studies to explore the role of severity and susceptibility in greater detail to clarify the importance, when other factors are present. For seasonal influenza the data suggest that women who believed that the disease could be harmful to their pregnancy or baby had four-times greater odds of being vaccinated than those who did not (OR 3.70, 95% CI 1.37-9.94) yet there was no evidence to suggest the belief in the risk of the disease generally (OR 1.56, 95% CI 0.88-2.76) or its ability to result in hospitalisation (OR 0.57, 95% CI 0.22-1.45) were related to vaccine uptake.”

Please see above in relation to Comment 23b) regarding definitions of “awareness/informed” as outlines in the appendix. It does not relate to severity or susceptibility and hence not included here. 

Comment 29: line 377 – your findings actually don’t support that messaging in vaccine campaigns should not focus on disease threat or hospitalisation for seasonal flu

Response: We agree with the reviewer that we should interpret with a greater degree of caution since lack of evidence does not equate to unimportance. However, it does suggest that messaging campaigns should be cautious of focusing on disease severity and susceptibility without further research analysing its relationship to messaging or absence of messaging on vaccine safety. 

Please see inclusion of all data and its interpretation on Page 17, Lines 402-407 of edited manuscript:

“For seasonal influenza, the data is inconclusive since women who believed that the disease could be harmful to their pregnancy or baby had four-times greater odds of being vaccinated than those who did not (OR 3.70, 95% CI 1.37-9.94) yet there was no evidence to suggest belief in the risk of the disease generally (OR 1.56, 95% CI 0.88-2.76) or its ability to result in hospitalisation (OR 0.57, 95% CI 0.22-1.45) were related to vaccine uptake.”

We have edited this statement as it now appears on Page 17, Lines 409-413: 

“This has important implications for public health communication strategies around maternal vaccination since campaigns particularly during an epidemic or influenza outbreak have centered around disease threat. Based on our findings we caution any communication approach which highlights only the threat of disease when publicising vaccination. We suggest this requires further review of the messaging strategies comparing those with and without explicit details of vaccine safety to the public and/or a discussion of disease threat with attention to language that might inadvertently promote fear [77].”

Comment 30: page 22 – line 397 – I don’t think I agree with this assertion for the above reasons

Response: We suggest modification, not removal of disease severity and susceptibility within this model. We recognize that there is inconclusive evidence but very strong evidence for the role of healthcare worker recommendation and the belief in vaccine harm. 

Comment 31: page 24 – line 441 – again I disagree here – rather than making them fear the disease, awareness and explanation of disease risk is important to weigh the perceived vaccine risks

Response: We agree with the reviewer that it is important to be informed and this is crucial to inform consenting procedures regarding any medical intervention, including vaccination. However, we are referring to the manner in which messages are communicated. Our point here is that if the focus of the conversation or message is on disease risk with minimal discussion around vaccine safety there will be decision conflict with potential fear of both leading to no vaccination. Whereas if disease risk is moderated against vaccine safety messaging this is more likely to be powerful based on our data. Informed consent it still paramount (informing regarding disease risk). We have included the following on Page 19, Lines 485-488: 

“Whilst it is essential that pregnant women are informed about the risks of the disease in order to be appropriately consented, the manner in which this is communicated should be evaluated. It appears that fear may be counter-productive.”

Comment 32: page 25 – line 472 – limitations - it is very unclear which studies you had no intention/uptake data, just intention or just uptake – if this is the case how can all your findings be presented in terms of vaccine uptake?

Response: The manuscript only refers to vaccine uptake data (our primary analysis). Our secondary analysis vaccine intention is included in the appendix. To make it easier for the reader all the data can be found in the Appendix Tables outlining which studies are for our 1) primary outcome (uptake) and 2) secondary outcome (intention). We focused on vaccine uptake as we believed this was a more objective measure of vaccination behaviour. 

Our first line of our discussion indicated this on Page 16, Line 365-367: 

“Despite the challenges of synthesizing an extensive and varied body of research, we have been able to quantify the relative effect size for a large number of specific beliefs and behaviours around maternal vaccination uptake.”

Additionally, on Pages 20, Lines 526-532: 

“Whilst we intended to compare results from the meta-analyses with vaccination status and intention to be vaccinated as outcomes, data were insufficient to draw meaningful comparisons. All the data for vaccine intention is found in the Appendix 23 p91. This is important since the majority of data available for pertussis vaccination in pregnancy focused on beliefs associated with vaccination intentions only [80, 87, 104, 105]. Whilst previous literature has shown intention to vaccinate can be a proxy for actual vaccination status, this may not always be the case with maternal vaccination and additional research is needed [67, 106].”

Comment 33: page 25 – line 495 – I agree that communication should focus on safety but that disease threat needs to be addressed as well – it is not one or the other and your data don’t actually support the disease threat for S flu was not significant in predicting vaccine “uptake”

Response: We agree with the reviewer that it is important to be informed and this is crucial to inform consenting procedures regarding any medical intervention, including vaccination. However, we are referring to the manner in which messages are communicated. Our point here is that if the focus of the conversation or message is on disease risk with minimal discussion around vaccine safety there will be decision conflict with potential fear of both leading to no vaccination. Whereas if disease risk is moderated against vaccine safety messaging this is more likely to be powerful based on our data. At the other extreme if vaccine safety messaging was highlighted without emphasis on disease severity or susceptibility our data suggests (but does not prove) this may still (even if counterintuitive) be powerful – however we would not recommend this strategy since informed consent it still paramount (informing regarding disease risk). We have included this on Page 19, Lines 585-588: 

“Whilst it is essential that pregnant women are informed about the risks of the disease in order to be appropriately consented, the manner in which this is communicated should be evaluated. It appears that fear may be counter-productive.”

Please see inclusion of all data and its interpretation on Page 17, Lines 402-407: 

“For seasonal influenza, the data is inconclusive since women who believed that the disease could be harmful to their pregnancy or baby had four-times greater odds of being vaccinated than those who did not (OR 3.70, 95% CI 1.37-9.94) yet there was no evidence to suggest belief in the risk of the disease generally (OR 1.56, 95% CI 0.88-2.76) or its ability to result in hospitalisation (OR 0.57, 95% CI 0.22-1.45) were related to in relation to vaccine uptake”.

Reviewer #1: In general, looking more closely at what is understood about barriers to improving uptake of maternal vaccination is a very important undertaking. This study has provided a comprehensive examination of what has been published on decision making around vaccination for 2 routinely recommended vaccines, as well as the pandemic H1N1 vaccine. I think the overall methodology is in keeping for systematic review protocol and the authors adhered to rigorous processes for distilling the data into a useful summary. 

Thank you. 

I did feel that there was a general lack of contextual information around the advancement of recommendations for influenza and pertussis-containing vaccines and how, at least in the US, we’ve observed shifts in the recommendation language in the study period being considered (1996-2018) for both influenza and Tdap vaccines, as well as the uptake of these vaccines. By including studies from such a broad study period, there have to be important differences based on the timing of the included publications. I don’t see any mention of this issue from the authors and feel that some discussion on this is warranted (more in Methods below).

Thank you - it is important to provide more context as you suggest – we have now included a paragraph in our introduction incorporating a short summary (see below). We have hopefully addressed this point in Comment 11

One difficulty it because the aim of the systematic review and meta-analysis was to perform a global analysis rather than a country specific analysis, it is difficult to provide detailed background for routine vaccination of seasonal influenza and pertussis vaccination in different years. For instance, looking specifically at pertussis, in the UK following the 2012 outbreak, pertussis vaccine was recommended routinely by HCW. However, this was instated by Israel in 2015, Australia in 2013, the USA in 2011. 

Page 3, Lines 66- 77 of the edited Introduction now state: 

“Historically, maternal tetanus vaccination was limited to areas of significant transmission. In areas where there is ongoing maternal to neonatal transmission of tetanus, two doses of TTCV (preferably Tetanus-diphtheria) are recommended in pregnancy in addition to Tdap or DTaP (for pertussis) and seasonal influenza vaccines.[2] Pertussis vaccination was limited to childhood however the resurgence of pertussis during outbreaks that disproportionately affected younger infants, led to national policy changes between 2011 and 2015, in countries such as the United Kingdom and the United States, that introduced routine maternal pertussis vaccination.[2-3] Similarly, the widespread influenza immunisation programs during the 2009 H1N1 pandemic resulted in public health bodies particularly in Europe, the United States and Australia introducing guidance to implement recommendations for routine annual seasonal influenza vaccination during the subsequent decade.”

The challenge with including studies from around the world is that the recommendations and barriers really do vary by country. 

Country specific analysis is important but a global perspective will help provide a broader analysis of factors that are common across location. Unfortunately, the data was not sufficient to stratify our results by Gross Domestic Product which we had planned to do from our pre-determined sensitivity analysis. 

My second overarching issue has to do with study samples and methods for determining eligibility in the included studies. Some methodologic process should be used to assess how populations were sampled and surveyed (or interviewed) to address whether there are potential biases that could be introduced.

See Comment 12. This is a great point and we have included the sample methods for included studies in the main summary table in the manuscript. The JBI critical appraisal tool including sample methods as a factor to assess each study, and so this was reflected in the JBI scores included in the Appendix. Additionally, this table provides the breakdown of sampling methods used in the 120 final included studies: 

Sample Method Count of Studies

Cluster 2

Convenience 45

Maximal variation 2

Not specified 18

Opportunistic 1

Purposive 24

Quota 1

Random 23

Stratified non-random 2

Stratified random 2

Total 120

Reviewer #2: Very well written. 

Thank you 

I don’t agree with the overall assertion that vaccine safety should be focused on at the expense of disease severity as your data don’t support this overall. My overall concern is that your interpretation of results data don’t clearly allow the reader to determine whether these factors are related to vaccine uptake or intention or neither and that suggestion to preferably focus on vaccine safety seems to be at the expense of addressing disease severity which is dangerous. I think this needs to be more balanced. Also this data seems to only be presented in Fig 2 for flu and not for pertussis? For overall clarity I think all these issues should be addressed

We hope that we have now provided a more moderated assertion regarding the importance of disease severity and susceptibility. We have now addressed all your comments and included all the data in the discussion to allow the reader a full view of the results and described these as “inconclusive” rather than unimportant. We agree that disease severity and susceptibility should be addressed with patients, and have adapted our manuscript to reflect that it is the communication and language used around these aspects since fear can result in a counter-intuitive response. However, it is paramount disease severity and susceptibility are kept within the consenting process of any patient. 

We have edited our abstract to state: Page 2, Lines 56-58:

“Public health campaigns which centre on the protectiveness and safety of a maternal vaccine rather than disease threat alone may prove beneficial.”

We hope we have addressed why the data included in the main table is on influenza (pandemic and seasonal) since the data extracted from the literature on pertussis in all in relation to our secondary outcome – vaccine intention. The results of which can be found in the Appendix. 

We hope we have now addressed these issues and thank you for your time in reviewing the manuscript and valuable contributions to enhance its message

---

## [Decision Letter · Decision Letter 1]

27 May 2020

PONE-D-20-05667R1

Factors that influence vaccination decision-making among pregnant women: a systematic review and meta-analysis

PLOS ONE

Dear Dr. Kilich,

Thank you for submitting your manuscript to PLOS ONE. After careful consideration, we feel that it has merit but does not fully meet PLOS ONE’s publication criteria as it currently stands. Therefore, we invite you to submit a revised version of the manuscript that addresses the last few minor points raised during the review process.

We look forward to receiving your revised manuscript.

Kind regards,

Ray Borrow, Ph.D., FRCPath

Academic Editor

PLOS ONE

Reviewers' comments:

Reviewer's Responses to Questions

**Comments to the Author**

1. If the authors have adequately addressed your comments raised in a previous round of review and you feel that this manuscript is now acceptable for publication, you may indicate that here to bypass the “Comments to the Author” section, enter your conflict of interest statement in the “Confidential to Editor” section, and submit your "Accept" recommendation.

Reviewer #1: All comments have been addressed

Reviewer #3: All comments have been addressed

2. Is the manuscript technically sound, and do the data support the conclusions?

Reviewer #1: Yes

Reviewer #3: Yes

3. Has the statistical analysis been performed appropriately and rigorously? 

Reviewer #1: Yes

Reviewer #3: I Don't Know

4. Have the authors made all data underlying the findings in their manuscript fully available?

Reviewer #1: Yes

Reviewer #3: (No Response)

5. Is the manuscript presented in an intelligible fashion and written in standard English?

Reviewer #1: Yes

Reviewer #3: Yes

6. Review Comments to the Author

Reviewer #1: The authors have done a very nice job of addressing all the submitted reviewer comments and provided appropriate and responsive modifications to the manuscript. The additional details and content that have been added are very helpful in providing important context to this review. The one comment I will make is in reference to the authors comment that there are no stated maternal vaccination targets available. While this is specific to the US, there is the Healthy People 2020 target of achieving influenza vaccination coverage of 80% among pregnant women. This is not presented as an ultimate target, but does suggest that there is a target goal. If the authors think this is useful, it might be worth using to help substantiate the idea that 70% coverage is considered "low".

Reviewer #3: Nicely written and important work, and concerns of the reviewers were sufficiently addressed.

Only minor comments:

- Please provide references in line 106

- Motivational interviewing techniques are strategies that can increase vaccine acceptance. Please include this strategy in your perspectives in the discussion.

7. PLOS authors have the option to publish the peer review history of their article (what does this mean?). If published, this will include your full peer review and any attached files.

Reviewer #1: No

Reviewer #3: No

---

## [Author Response · Author response to Decision Letter 1]

30 May 2020

Dear Editor,

Thank you for the constructive comments from the reviewers on our manuscript. We have addressed these two minor comments (below) in turn and the manuscript has been amended accordingly. 

Thank you, again, for considering our revised manuscript. 

Kind Regards,

Dr Eliz Kilich

Corresponding Author

Reviewer 1 Comment 1: The authors have done a very nice job of addressing all the submitted reviewer comments and provided appropriate and responsive modifications to the manuscript. The additional details and content that have been added are very helpful in providing important context to this review. The one comment I will make is in reference to the authors comment that there are no stated maternal vaccination targets available. While this is specific to the US, there is the Healthy People 2020 target of achieving influenza vaccination coverage of 80% among pregnant women. This is not presented as an ultimate target, but does suggest that there is a target goal. If the authors think this is useful, it might be worth using to help substantiate the idea that 70% coverage is considered "low".

Response: Thank you, this is a very helpful suggestion. We have added the statement on Line 77-78 and adapted our references: 

“The United States Healthy People 2020 campaign sets a target to achieve influenza vaccination coverage of 80% among pregnant women [4].”

Reviewer 3 Comment 2: Nicely written and important work, and concerns of the reviewers were sufficiently addressed. Only minor comments: a) Please provide references in line 106. b) Motivational interviewing techniques are strategies that can increase vaccine acceptance. Please include this strategy in your perspectives in the discussion.

Response: 

a) References have now been provided for line 106. References [11-15] are inserted. 

b) To include interventions such as motivational interviewing we have adapted Line 554-556 and adapted our references:

“Interventions recommended to improve maternal vaccination uptake have ranged from text reminders for prospective mothers to educational videos and motivational interviewing techniques for HCPs [110-112].”

[110] Gagneur A. Motivational interviewing: A powerful tool to address vaccine hesitancy. Can Commun Dis Rep. 2020, 46(4): 93-97. doi:10.14745/ccdr.v46i04a06.

---

## [Editor Report · Decision Letter 2]

3 Jun 2020

Factors that influence vaccination decision-making among pregnant women: a systematic review and meta-analysis

PONE-D-20-05667R2

Dear Dr. Kilich,

We’re pleased to inform you that your manuscript has been judged scientifically suitable for publication and will be formally accepted for publication once it meets all outstanding technical requirements.

Kind regards,

Ray Borrow, Ph.D., FRCPath

Academic Editor

PLOS ONE
---

## [Editor Report · Acceptance letter]

23 Jun 2020

PONE-D-20-05667R2 

Factors that influence vaccination decision-making among pregnant women: a systematic review and meta-analysis 

Dear Dr. Kilich:

I'm pleased to inform you that your manuscript has been deemed suitable for publication in PLOS ONE. Congratulations! Your manuscript is now with our production department. 

Kind regards, 

on behalf of

Prof. Ray Borrow 

Academic Editor

PLOS ONE